# Implementation of a High-Sensitivity Global Navigation Satellite System Receiver on a System-on-Chip Field-Programmable Gate Array Platform

**DOI:** 10.3390/s24051416

**Published:** 2024-02-22

**Authors:** Marc Majoral, Javier Arribas, Carles Fernández-Prades

**Affiliations:** Centre Tecnològic de Telecomunicacions de Catalunya (CTTC/CERCA), Parc Mediterrani de la Tecnologia, Building B4, Av. Carl Friedrich Gauss 7, 08860 Castelldefels, Spain; jarribas@cttc.es (J.A.); carles.fernandez@cttc.es (C.F.-P.)

**Keywords:** GNSS, FPGA, system on chip, SoC-FPGA, high-sensitivity GNSS receiver, software-defined radio

## Abstract

This paper presents the design, proof-of-concept implementation, and preliminary performance assessment of an affordable real-time High-Sensitivity (HS) Global Navigation Satellite System (GNSS) receiver. Specifically tailored to capture and track weak Galileo E1b/c signals, this receiver aims to support research endeavors focused on advancing GNSS signal processing algorithms, particularly in scenarios characterized by pronounced signal attenuation. Leveraging System-on-Chip Field-Programmable Gate Array (SoC-FPGA) technology, this design merges the adaptability of Software Defined Radio (SDR) concepts with the the robust hardware processing capabilities of FPGAs. This innovative approach enhances power efficiency compared to conventional designs relying on general-purpose processors, thereby facilitating the development of embedded software-defined receivers. Within this architecture, we implemented a modular GNSS baseband processing engine, offering a versatile platform for the integration of novel algorithms. The proposed receiver undergoes testing with live signals, showcasing its capability to process GNSS signals even in challenging scenarios with a carrier-to-noise density ratio ($C/N_{0}$) as low as 20 dB-Hz, while delivering navigation solutions. This work contributes to the advancement of low-cost, high-sensitivity GNSS receivers, providing a valuable tool for researchers engaged in the development, testing, and validation of experimental GNSS signal processing techniques.

## 1. Introduction

GNSS technology allows us to accurately know our location in real time and in open-sky environments. Due to the success achieved by this technology, there is an increasing demand for navigation in signal-challenged environments such as foliage canopy, urban canyons, and indoor scenarios [1]. However, these environments present great difficulties for GNSS receivers, as the received signals are severely degraded due to the presence of obstacles in the propagation path between the satellites and the receiver. Real-world usage of a GNSS receiver in indoor scenarios presents various operational challenges. The standard outdoor working conditions, characterized by nominal $C/N_{0}$ values of typically ≥44 dB-Hz, differ significantly from the indoor environment, where signal levels are reduced after passing through roofs, walls, and windows [2,3]. Indoor scenarios lead to inferior signal detection and degrade the quality of measurements due to higher noise and multipath effects. The major degradation that the indoor receivers have to cope with is high signal attenuation [4]. The signal reception is influenced by building materials and the receiver’s location, resulting in attenuation losses of approximately 10 to 20 dB in soft-indoor scenarios, 20 to 35 dB in indoor scenarios, and exceeding 35 dB in deep indoor scenarios [3]. The terms ‘soft indoor’, ‘indoor’, and ‘deep indoor’ describe varying degrees of GNSS satellite signal obstruction caused by construction materials or vegetation. Soft indoor scenarios occur when GNSS signals are weakened by partial blockages, reflections, and diffractions as they navigate through or around buildings and other obstacles. These scenarios still provide some level of GNSS signal availability. Indoor and deep indoor scenarios involve more significant signal blockage, resulting in greatly reduced signal strength [3,5].

As a response to this challenge, HS techniques have been developed for GNSS receivers [6]. HS-GNSS receivers enable improved acquisition and tracking capabilities in degraded signal environments, albeit at the cost of increased complexity in terms of computational load [7]. Existing solutions that provide high accuracy and sensitivity are often constrained by their elevated energy consumption [6]. Methods to alleviate power usage encompass snapshot solutions, cloud-based receivers, and assisted (A-GNSS) receivers [2,6].

Snapshot receivers obtain the navigation solutions by analyzing a brief segment of the received satellite signal. This involves sampling times on the order of a few milliseconds. The distinctive feature of a snapshot receiver lies in its capacity to function effectively within these brief signal sampling intervals. This characteristic makes it well-suited for a diverse range of positioning applications where the constraint of energy usage poses a significant challenge in embracing traditional GNSS solutions [8].

Cloud-based GNSS receivers leverage shared computational resources. Computational tasks typically carried out on-chip are migrated to a cloud server with the objective of enhancing the sensor’s battery lifetime without compromising the performance [9,10]. The cloud-based receiver can be set up in a manner where solely the signal capture circuitry is employed to temporarily store the digital samples. Signal processing can be deferred until the digital samples can be transmitted, without impacting the device’s battery life (e.g., during battery recharge), to a distant cloud server [6].

A-GNSS solutions enhance the performance of standard receivers by delivering information, using an alternate communication pathway, that the receiver would typically acquire directly from the satellites. Assistance reduces the time and information dependency on satellites for obtaining navigation solutions. Consequently, the A-GNSS receiver can swiftly make measurements from the satellites, even with weaker signals, surpassing the capabilities of an unassisted receiver [2].

HS techniques are primarily developed to capture faint signals. This is usually achieved by coherently accumulating signal samples over an extended period, known as the coherent integration (CI) [6]. However, several critical factors impose constraints on the maximum CI time, including the presence of unknown data bit transitions, residual frequency errors in the received signal, and the existence of phase noise due to receiver oscillator instabilities [3].

Sign reversals within the CI window may occur due to unknown data bit transitions in the received signal, potentially causing partial or complete cancellation of correlation power. The impact of these unknown bit transitions can be mitigated by using pilot signals [6]. A pilot signal is transmitted within the Galileo signals (E1, E5a, E5b, E6) and modernized Global Positioning System (GPS) signals (L1C, L2C, L5) to improve the signal acquisition and tracking [3]. Both the data and the pilot signals are modulated with a unique code to distinguish them from other signals. This code is a pseudo-random noise (PRN) sequence known as a spreading code. Several GNSS signals build a long spreading sequence in a tiered manner whereby a secondary code sequence is used to modify successive repetitions of a primary code.

Residual frequency errors impact the received signal after Doppler wipe-off, potentially leading to signal cancellation if the CI exceeds a certain time duration. A smaller residual frequency shift allows for an extended CI before the Signal-to-Noise Ratio (SNR) gain diminishes due to phase wrapping. This phenomenon introduces a trade-off between SNR gain and computational load, as minimizing the residual frequency shift necessitates a finer Doppler search, demanding increased computational operations [3].

The quality of the user receiver clock also limits the CI time. At present, temperature-compensated crystal oscillators (TCXOs) and oven-controlled crystal oscillators (OCXOs) are the most commonly utilized types of clocks. As per the simulations outlined in [7,11,12], the TCXO restricts the CI interval to approximately 100 ms, whereas the OCXO enables the utilization of CI times extending up to 1 s. The sensitivity of the receiver is influenced by this fact. Consumer-grade receivers commonly utilize TCXO clocks, while OCXO clocks are typically reserved for professional applications. The higher cost of OCXO clocks, as opposed to TCXO clocks, restricts their widespread adoption in mass-market GNSS receivers [6].

For practical considerations, the CI time cannot be indefinitely extended due to the reasons mentioned above. Consequently, the only recourse is to employ a combination of coherent and non-coherent integration (NCI) to prolong the overall integration time and enhance receiver sensitivity [3,7]. More specifically, the acquisition begins with coherent correlation, followed by non-coherent accumulation of outputs from multiple coherent correlations through nonlinear operations. Non-coherent accumulation is commonly achieved through the application of Post-Detection Integration (PDI) techniques. These techniques address the constraints associated with coherent accumulation and enable the receiver to capture satellites even in conditions of extremely low $C/N_{0}$ [7]. However, the NCI time cannot be increased without bounds either, due to limited Doppler estimation accuracy, receiver clock drift instabilities, and the relative movement between the satellites and the receiver [3].

Any errors in determining the Doppler frequency can affect the baseband signal, akin to a mismatch between the chip duration of the received signal and the local code replica. If this mismatch extends over a significant integration interval, it causes blurring in the overall correlation, introducing bias in the final estimation of the code phase [3]. The duration of the NCI also remains limited due to local clock drift instabilities, unless methods to estimate the clock dynamics are implemented [3]. In addition, processing the received signal over the total integration time yields a single Doppler frequency and code phase estimation for each satellite. However, the accuracy of this time-delay estimation is limited by the dynamic motion of both the satellites and the receiver during the correlation time [3].

Acquiring weak GNSS signals requires significant computational effort due to the need for extended integration. The most time-consuming operation during the acquisition phase is correlating the input signal with a locally generated replica. This process involves multiplying thousands of samples every millisecond for each tested code phase, Doppler frequency, and satellite [3]. Consequently, developing a real-time High Sensitivity GNSS (HS-GNSS) receiver presents a major challenge. Several strategies have been explored to reduce the computational demands of GNSS signal acquisition algorithms, with some examples provided below.

The Double Block Zero Padding (DBZP) approach, as detailed in [13], employs partial correlations of short durations (typically a few tens of chips) to calculate the CI using multiple small Fast Fourier Transforms (FFTs) instead of a single large FFT. Another strategy, proposed by [14], focuses on the joint acquisition of the in-phase and quadrature components of the GNSS signal. It utilizes a cascaded FFT technique for acquiring both the primary and secondary codes. Refs. [15,16] introduced two algorithms designed to extend the CI time to multiples of the data bit interval, without requiring prior knowledge of the transmitted modulated data. These algorithms are named circular correlation with multiple data bits (CCMDB) and modified DBZP. Both approaches aim to predict the most likely data bit combination, using this prediction to adjust the data signs accordingly. The CCMDB algorithm reduces computational demands by initially setting a short CI time, then gradually increasing it, selectively eliminating Doppler frequency bins based on their likelihood. The modified DBZP technique offers a means to account for the Doppler effect over the PRN code’s duration with only a slight increase in processing needs. Another method, utilizing a double-FFT algorithm [3], performs CI over a bit period without prior knowledge of bit transitions. This approach applies a sliding FFT window equal to one bit’s duration, facilitating the detection of bit transitions and enabling coherent integration of correlation outputs.

### 1.1. Motivation

GNSS-based applications are increasingly expanding their reach to encompass more demanding scenarios, including urban and light-indoor environments. In areas with weak or compromised signals, they improve the availability of satellite signals [1,4]. This capability enhances the overall reliability and performance of GNSS receivers in diverse settings. This is significant for various applications, including location-based services (LBSs) and emergency response [17]. Operating effectively in conditions where traditional receivers may struggle, HS-GNSS receivers provide extended coverage in marginal signal conditions. The heightened need for navigation in challenging signal environments has led to a growing focus on handling faint signals [1]. This shift is motivating the development of receivers that operate in these challenging scenarios [6].

The development of HS-GNSS receiver technology necessitates the implementation of non-standard features and a comprehensive description of the signal processing path from the antenna to the computation of the desired GNSS products. However, developing prototypes using off-the-shelf GNSS receivers poses difficulties. Researchers must contend with the increasing complexity and integration level of GNSS integrated circuits [18,19]. These systems do not offer an exact model of how the desired measurements are obtained, and they have limited reprogrammability.

In light of these challenges, the primary motivation for the research presented in this paper is to develop a programmable, adaptable, and portable HS-GNSS receiver prototype. This prototype is aimed at researching experimental algorithms, particularly those focused on positioning in weak signal conditions.

### 1.2. Contributions

In this paper, the authors designed, implemented, and tested a HS-GNSS receiver. The proposed receiver implements two operating modes: high-sensitivity mode and normal-sensitivity mode. When operating in high-sensitivity mode, the receiver is capable of acquiring and tracking Galileo E1b/c signals with a $C/N_{0}$ down to 20 dB-Hz (equivalent $C/N_{0}$ observed at the post-correlation level), enabling the derivation of navigation solutions. On the other side, when operating in normal-sensitivity mode, the receiver processes GPS L1 C/A, Galileo E1b/c, GPS L5, and Galileo E5a signals with an acquisition sensitivity of approximately 37 dB-Hz. While the ideal scenario involves acquiring and tracking both GPS and Galileo signals in high-sensitivity mode, the initial concept demonstrator presented in this paper is designed to showcase high-sensitivity mode only for Galileo E1b/c signals.

To enhance the availability of satellite signals, the receiver is capable of processing Galileo E1 b/c signals in high-sensitivity mode, while simultaneously processing GPS L1 C/A, GPS L5, and Galileo E5a signals in normal-sensitivity mode. This approach implements a dual-band, multi-GNSS receiver centered at 1176.45 MHz and 1575.42 MHz.

When operating in high-sensitivity mode, the receiver uses assistance to speed up the acquisition of weak Galileo E1b/c signals and ultimately to decrease the time-to-first-fix (TTFF).

The proposed design is based on the SoC-FPGA receiver architecture introduced in [20,21]. In the presented work, the authors upgraded this architecture and developed a concept demonstrator implementing high sensitivity mode. This enhancement enables the processing of weak Galileo E1b/c signals, thereby obtaining navigation solutions from these signals.

This paper focuses on the high-sensitivity capabilities of the proposed receiver. A brief description of the SoC-FPGA receiver architecture is also provided. A more detailed description of the SoC-FPGA receiver architecture and design methodology can be found in [21].

### 1.3. Organization of the Paper

The remainder of this paper is structured as follows: Section 2 reviews previous work investigating ways to increase the sensitivity of GNSS receivers using a combination of CI and NCI, employing PDI techniques. Additionally, it discusses previous work investigating ways to implement these techniques in receivers based on FPGAs. Section 3 describes the design of the proposed receiver. The design description mainly focuses on the newly introduced features that enable the receiver to process severely attenuated signals and obtain navigation solutions. Section 4 reports the performance test results. Finally, Section 5 presents the conclusion and directions for future work.

## 2. Literature Review

Several publications explore the application of CI and PDI techniques to increase the sensitivity of GNSS receivers. Ref. [11] investigates and assesses the performance of several PDI techniques, including Non-coherent Post-Detection Integration (NPDI), Differential Post-Detection Integration (DPDI), Generalized Post-Detection Integration Truncated (GPDIT), Squaring Detector (SD), and GPDITSD, which employs the GPDIT strategy and the SD. The evaluation of these techniques involves the use of three types of clocks: a TCXO, a Chip Scale Atomic Clock (CSAC), and an OCXO. In [22], an analysis is performed to identify the most effective PDI technique in the presence of various impairments like data bit transitions and frequency offset. Additionally, [12] conducts a comprehensive investigation into current PDI techniques, specifically addressing the impact of phase noise originating from two different clocks: a TCXO and an OCXO. Ultimately, the PhD thesis [7] consolidates comprehensive insights and benchmarks the most relevant PDI techniques for acquiring GNSS signals under the challenging conditions mentioned above (phase noise, frequency offset, and the presence of data bits). According to the simulation results presented in [7], a receiver equipped with a TCXO can reliably acquire GNSS signals at a $C/N_{0}$ as low as 20 dB-Hz. This achievement is possible by employing the GPDIT strategy, using a CI time of 100 ms, and allowing for a maximum of seven non-coherent combinations. These conditions result in a total integration time of 700 ms. The GPDIT strategy is not robust against data bit transitions, necessitating that the receiver either eliminates the modulated data or utilizes pilot signals. This approach is implementable in the FPGA and facilitates the acquisition of GNSS signals in soft indoor scenarios. For this reason, the HS-GNSS receiver prototype introduced in this paper adopts the GPDIT strategy and acquires the pilot signals.

The proposed receiver employs the Parallel Code Phase Search (PCPS) algorithm to achieve CI by performing circular cross-correlation in the frequency domain between the received signal and a locally generated replica of the satellite’s PRN code [23]. The PhD thesis [24] explores methods to optimize the implementation of the PCPS algorithm for HS-GNSS receivers, aiming for reduced complexity without incurring implementation losses. The detection of weak signals demands extended CI times. Consequently, the computation of the circular cross-correlations accounts for the presence of both primary and secondary codes in the pilot signals and requires the use of large FFTs and inverse FFTs (IFFTs). This thesis investigates the algorithm’s implementation in FPGAs, where FPGA vendors provide FFT and IFFT hardware accelerators as Intellectual Property (IP) cores. These accelerators typically have limitations on transform lengths, requiring them to be powers of two. As a result, the optimization of the PCPS algorithm is driven by the need to efficiently compute large FFTs using small power-of-two length FFTs and the need to minimize resource usage and algorithm latency. Ref. [24] shows ways to compute an *N*-point FFT using a combination of two FFTs, each of $\frac{N}{2}$ points. One way to do it is by separating the input samples by parity and the output samples by section. This method can be generalized to compute an *N*-point FFT using *P* FFTs, each of $\frac{N}{P}$ points. The same method is applicable to the computation of IFFTs. The receiver presented in this paper adopts this approach. This method enables the efficient implementation of the CI in the FPGA using a multi-channel FFT, with an optimized pipeline scheme that minimizes latency, albeit with a small increase in mathematical operations compared to the computation of a single large FFT. This algorithm does not require complex branch decisions, as it relies on a streamlined and efficient process, resulting in faster execution.

Ref. [24] also illustrates that selecting the optimal implementation of an algorithm for an FPGA entails several trade-offs, including the balance between speed and FPGA resource utilization, managing power consumption while optimizing performance, and algorithm complexity versus available FPGA resources.

## 3. System Design

### 3.1. High Sensitivity GNSS Receiver Architecture

The HS-GNSS receiver outlined in this paper is implemented using the SoC-FPGA receiver architecture presented in [21]. Comprehensive details of the proposed architecture can be found in the same reference, with only a brief overview provided here. This section primarily focuses on detailing the design aspects of the receiver that contribute to its high-sensitivity capabilities.

Figure 1 shows a block diagram of the receiver architecture. The receiver is implemented in an SoC FPGA, which is internally divided into Programmable Logic (PL) and a Processing System (PS). The PL consists of FPGA logic, while the PS houses an embedded processor. The FPGA receives incoming samples from the Radio Frequency Front End (RFFE) and implements a hardware accelerator for the acquisition process, as well as multiple hardware accelerators for the tracking multicorrelation tasks. The embedded processor executes GNSS-SDR, a widely recognized open-source software-defined GNSS receiver available online [25,26]. GNSS-SDR operates on an embedded GNU/Linux Operating System (OS) and controls the hardware accelerators in real-time. It implements various receiver channels, which encapsulate acquisition, tracking, and telemetry decoder blocks. The acquisition and tracking blocks in the software receiver offload part of the acquisition and tracking processes to the FPGA, enabling the execution of the GNSS engine in an embedded processor. GNSS-SDR handles the hardware accelerators in the FPGA using memory-mapped registers and interrupt request lines going from the hardware accelerators to the embedded processor. Additionally, GNSS-SDR computes the observables and the Position, Velocity, and Time (PVT) solutions. The observables include pseudorange, carrier phase, and Doppler shifts. The SoC FPGA architecture and design methodology is explained in detail in [21]. The GNSS-SDR software receiver is explained in detail in [25,26].

The proposed HS-GNSS receiver prototype is based on Advanced Micro Devices’ (AMD) ZCU102 development board [27], featuring an AMD Zynq UltraScale+ XCZU9EG-2FFVB1156 All-Programmable Multi-Processor SoC (MPSoC) [28]. A block diagram of the concept demonstrator is shown in Figure 2. The XCZU9EG MPSoC features Programmable Logic (PL) equipped with 600k logic cells and 2520 DSP slices and a Processing System (PS) that houses a Quad ARM Cortex-A53 MultiProcessor Core (MPCore) running at 1.3 GHz. It also includes several peripheral interfaces, such as a Local Area Network (LAN) interface, enabling the remote control of the receiver. The receiver incorporates an Analog Devices AD-FMCOMMS5-EBZ analog front-end [29], connected to the ZCU102 evaluation board. The AD-FMCOMMS5-EBZ includes two AD9361 Radio Frequency (RF) transceivers ([30]) covering the full 6 GHz range. Specifically, one RF transceiver is tuned to the E1/L1 frequency band, and the other is tuned to the E5a/L5 frequency band. This platform enables the implementation of a Dual-Band GNSS receiver, offering end-to-end functionality from the RF output of the antenna to real-time generation of navigation products on a portable board. The receiver uses a Tallysman TW8825 antenna ([31]), which provides dual-band GPS L1/L5, GLONASS G1, Galileo E1/E5a, and BeiDou B1 coverage.

The local oscillator included with the AD-FMCOMMS5-EBZ board, an RXO3225M IC crystal by Rakon clocked at 40 MHz, has a stability of ±25 parts per million (ppm), which is insufficient for GNSS signal processing. Therefore, an external oscillator is needed. The proposed receiver implementation was tested using two external oscillators: a TCXO and an OCXO. The selected TCXO was the AST3TQ-50 by Abracon, in its 40 MHz version [32]. This device features a frequency stability of ±50 parts per billion (ppb) over a temperature range of −40 °C to +85 °C, making it well-suited for GNSS applications. The chosen OCXO was an ECOC-2522 by ECS [33], operating at 40 MHz, with a frequency stability of ±10 ppb.

Figure 3 illustrates the architecture of the FPGA. The FPGA incorporates the above mentioned sample conditioning and buffering, as well as the high-sensitivity acquisition and tracking multicorrelator hardware accelerators. Additionally, it features an interface to the processing system (PS/PL interface). The sample conditioning and buffering incorporates dynamic bit selection in each frequency band. This process dynamically selects the most significant bits from the incoming signal samples, mapping their dynamic range to the sample quantization in the hardware accelerators. The hardware accelerators are implemented in the form of reusable Intellectual Property (IP) cores, which can be targeted at many variants of FPGAs. The PS/PL interface uses the Advanced Microcontroller Bus Architecture (AMBA) Advanced eXtensible Interface (AXI4) protocol specification [34].

The incoming GNSS signal is captured by the A/D converters and then transferred to the FPGA buffers. These buffers feed the various hardware accelerators for real-time processing. The receiver comprises one high-sensitivity acquisition FPGA IP core and 48 tracking multicorrelator FPGA IP cores, distributed as follows: 12 tracking IP cores for GPS L1 C/A, 12 for Galileo E1b/c, 12 for GPS L5, and 12 for Galileo E5a. This configuration enables the receiver to process GPS L1 C/A, Galileo E1b/c, GPS L5, and Galileo E5a signals in any combination of GNSS signals and up to 12 channels per signal—allowing for a maximum of 48 channels in a dual-frequency E1/L1 and L5/E5a setup. The input of the acquisition hardware accelerator is preceded by a downsampling filter in the L1/E1 band. In this way, the receiver adjusts its sampling frequency downward for capturing GNSS signals in the E1/L1 frequency band, and, subsequently, it switches to a higher sampling frequency for tracking these signals. This configuration is in accordance with the fact that the signal detection is not significantly benefited from operating within a bandwidth exceeding that necessary to capture the main lobe of the received signals, which is approximately 2 MHz for GPS L1 C/A signals and 4 MHz for Galileo E1b/c signals, due to the increased noise in the captured signal [35].

A Direct Memory Access (DMA) in the FPGA can be employed for processing recorded signals. Alternatively, the DMA can be used to capture snapshots of the received signals and transmit them to an external device via the PS and the LAN.

To integrate the high-sensitivity capabilities, significant modifications were made to the SoC-FPGA receiver architecture presented in [21]. These modifications included the implementation of a new acquisition hardware accelerator in the FPGA and modifications to the GNSS-SDR software receiver.

The new acquisition hardware accelerator is designed to perform extended CI, enabling the detection of weak signals in real-time. It is named the ‘high-sensitivity acquisition hardware accelerator’ because it enables the receiver to acquire signals in both standard and high-sensitivity modes, unlike the standard acquisition hardware accelerator presented in [21], which can only operate in normal sensitivity mode.

The high-sensitivity acquisition hardware accelerator necessitates temporary storage of received signal samples to compute the CI. To facilitate this, the acquisition is directly connected to a Double Data Rate 4 (DDR4) memory controller implemented in the FPGA. The memory controller, in turn, is connected to an DDR4 memory component on the ZCU102 board, which is located outside the SoC FPGA, but directly accessible by the FPGA itself. This memory component is called ‘PL memory’ because it is directly connected to the Programmable Logic (the FPGA). By utilizing this external memory, the acquisition hardware accelerator can efficiently store received signals temporarily and compute the CI, bypassing the PS. The PL memory is also accessible from the embedded processor, allowing the processor to read the results of the CI performed by the hardware accelerator. The PS is also connected to another DDR4 memory component supporting the embedded OS and used for program execution in the embedded processor. The memory component connected to the processing system is not illustrated in Figure 3.

To implement the high-sensitivity capabilities, the following modifications were applied to the GNSS-SDR software receiver: a new acquisition block was created in GNSS-SDR. This block processes the results computed by the new high-sensitivity acquisition hardware accelerator and conducts part of the acquisition algorithm in the embedded processor. GNSS-SDR was also modified to utilize assistance data for performing Doppler prediction, narrowing down the Doppler search space during acquisition. Moreover, GNSS-SDR was updated for tracking severely attenuated signals. Finally, GNSS-SDR was extended to use assistance data for obtaining navigation solutions in situations where the navigation message of weak GNSS signals cannot be reliably demodulated due to the presence of noise. With all these enhancements, the proposed HS-GNSS receiver produces PVT solutions in the presence of weak signals and in real-time. These features are elaborated further in the subsections below.

Figure 4 shows a picture of the HS-GNSS Receiver prototype. The receiver has been fully constructed with Commercial Off-the-Shelf (COTS) components. Incorporating the ZCU102 and AD-FMCOMMS5-EBZ development boards results in a sizable design. The ZCU102 board measures 23.749 cm × 24.384 cm, and the analog front-end measures 14 cm × 9 cm. Despite these dimensions, the receiver remains portable. For instance, it can be placed in backpacks to facilitate testing in soft indoor environments. The ZCU102 includes several components that are unnecessary for the implementation of the HS-GNSS receiver. Therefore, the size of the receiver could be potentially reduced by using a custom-designed Printed Circuit Board (PCB).

### 3.2. Receiver Operating Modes

The proposed concept demonstrator implements two operating modes: high-sensitivity mode and normal-sensitivity mode. High-sensitivity mode enables the processing of weak Galileo E1b/c signals with a $C/N_{0}$ down to 20 dB-Hz. Normal-sensitivity mode enables the processing of both Galileo signals and GPS signals received at nominal $C/N_{0}$ levels. The default configuration of the receiver is set to operate in high-sensitivity mode for Galileo E1b/c signals, and concurrently in normal-sensitivity mode for Galileo E5a, GPS L1 C/A, and GPS L5 signals, as illustrated in Table 1. This dual-mode setup is designed to enhance receiver availability whenever possible, given that high-sensitivity mode is currently only supported for Galileo E1b/c signals.

### 3.3. Assistance Data

The proposed demonstrator utilizes GNSS assistance data to process weak Galileo E1b/c signals. This assistance data is provided in various forms, including a reference date and time, a reference user location, Galileo ephemeris data, Galileo ionospheric data, and the Galileo Coordinated Universal Time (UTC) model, as detailed in Table 2. The receiver obtains the current date and time from the embedded GNU/Linux OS clock, retrieves the reference user location from the GNSS-SDR configuration file, and accesses the Galileo ephemeris data, ionospheric data, and UTC model from Extensible Markup Language (XML) files. These XML files, which contain data about the visible satellites, can be generated by the receiver itself by placing it outdoors and operating it in normal-sensitivity mode.

Table 3 outlines the purpose of the assistance data and specifies the data that is utilized for each designated task. The receiver uses the reference date and time, the reference user location, and the assistance ephemeris data to estimate the Doppler frequency of the received signals and reduce the Doppler search space during acquisition. The reference date and time are also utilized to estimate the transmitted Time of the Week (TOW) in the navigation messages. This estimation is needed because the TOW is essential for performing GNSS basic measurements. However, when tracking severely attenuated signals, the receiver may struggle to reliably demodulate the TOW field of Galileo E1b/c navigation messages due to noise-induced bit errors [36,37]. Additionally, the assistance ephemeris data, ionospheric data and UTC model are utilized to compute the PVT when the receiver cannot reliably demodulate the navigation messages due to the presence of bit errors.

The Doppler prediction, the TOW estimation, and the computation of the navigation solutions are explained in more detail in Section 3.5, Section 3.8, and Section 3.9, respectively.

### 3.4. Acquisition in High-Sensitivity Mode

#### 3.4.1. Theory of Operation

The acquisition process makes it possible to identify faint GNSS signals. Upon detecting a signal, the acquisition offers an estimation of both the code delay and Doppler frequency sufficiently accurate to initiate the tracking loops. The acquisition process is a two-dimensional operation, involving the correlation of the received signal $x_{IN}\left[n\right]$ with a local replica of the transmitted signal d[n] across various trial values of Doppler frequency $f_{d}$ and time delay τ. The circular correlation between $x_{IN}\left[n\right]$ and d[n] is referred to as the Cross Ambiguity Function (CAF). If we assume the absence of bit transitions in the received signal, the CAF can be expressed using Equation (Equation 1). In this equation, *N* represents the number of samples integrated coherently, and $T_{s}$ denotes the sampling period.
(1)$$R_{xd}\left(\tau ,f_{d}\right)=\frac{1}{N}\sum_{n=0}^{N-1}x_{IN}\left[n\right]d\left[nT_{s}-\tau \right]e^{-j2\pi f_{d}nT_{s}}$$

PDI techniques can be employed to overcome the limitation of extending CI indefinitely. These techniques involve combining multiple consecutive CAFs through a non-linear function, as depicted in Equation (Equation 2) [7]. In this equation, $Z_{X}$ is the result of the NCI obtained using PDI techniques, $R_{k}$ represents the *k*-th consecutive CAF, $N_{nc}$ denotes the number of non-coherent combinations, $f_{d}$ is the Doppler frequency, and τ is the time delay.
(2)$$Z_{X}=f\left(\sum_{k=1}^{N_{nc}}R_{k}\left(\tau ,f_{d}\right)\right)$$

When operating in high-sensitivity mode, the receiver described in this paper employs the GPDIT strategy [7] to acquire highly attenuated GNSS signals in real-time. The GPDIT strategy, illustrated in Equation (Equation 3), involves determining $Z_{\mathrm{GPDIT}}$ as the outcome of the NCI.
(3)$$\begin{array}{c} Z_{\mathrm{GPDIT}}=\sum_{k=1}^{N_{nc}}|R_{k}\left(\tau ,f_{d}\right){|}^{2}+2\left|\sum_{k=2}^{N_{nc}}R_{k}\left(\tau ,f_{d}\right)R_{k-1}^{*}\left(\tau ,f_{d}\right)\right| \end{array}$$

The GPDIT strategy is chosen for its effectiveness, as highlighted in [7]. This technique is deemed the most suitable in the presence of a frequency offset and demonstrates reasonable performance even in the face of phase noise resulting from the use of a TCXO. According to the findings in [7], GNSS signals with a $C/N_{0}$ as low as 20 dB-Hz can be acquired, while maintaining a probability of false alarm $P_{fa}<0.01$ and a probability of correct detection $P_{d}∼0.8$. This is achieved through CI over a time span of 100 ms and employing GPDIT with up to $N_{nc}=7$ non-coherent combinations. These parameters ensure a reliable acquisition of weak GNSS signals. While the GPDIT technique experiences significant degradation in the presence of data bits in the received signals [7], it proves effective when applied to pilot signals devoid of data and featuring predictable pilot codes. For this reason, the proposed high-sensitivity acquisition process acquires the Galileo pilot signals.

For a stationary GNSS receiver the maximum Doppler frequency shift in the E1/L1 frequency band is around ±5 kHz. The acquisition Doppler search step is chosen considering a trade-off between computational load at the receiver and accuracy in the estimation of the received Doppler frequency. A typical value for the Doppler search step is $f_{st}=\frac{1}{2T_{CI}}$, where $T_{CI}$ is the CI time [7]. When $T_{CI}$ is set to 100 ms, this formula yields a Doppler search step of 5 Hz. As a result, the acquisition must conduct a frequency sweep across all potential carrier frequencies within a ±5 kHz range, in 5 Hz increments. This process involves a vast number of combinations, leading to a significant computational burden. In practice, the use of assistance data and Doppler prediction reduces the acquisition Doppler search space and consequently the acquisition latency.

#### 3.4.2. Computation of Large FFTs in the FPGA

When working in high-sensitivity mode, the acquisition hardware accelerator computes the circular cross-correlation between the received signals and a local replica of the Galileo tiered codes over a time span of 100 ms. The circular cross-correlation is computed in the frequency domain, enabling efficient implementation within parallel processing architectures. This approach is particularly advantageous for applications requiring real-time signal processing. Many HS-GNSS receivers compute the CAF in the frequency domain using FFT-based techniques for their computational efficiency [7].

Performing circular correlations over a duration of 100 ms requires the use of very large FFTs, with sizes on the order of at least several hundred thousand samples. However, FPGA FFT IP cores are limited in length, requiring the FFT length to be a power of two. For this reason, the proposed FPGA implementation computes large FFTs by combining multiple smaller FFTs with complex exponentials. This approach is extended to IFFTs, following a similar procedure. To elaborate, this approach can be explained as follows: a large *N*-point Discrete Fourier Transform (DFT) can be computed by utilizing two smaller DFTs, each of $\frac{N}{2}$ points. This is achieved by separating the input sequence into even samples ($x_{2n}$) and odd samples ($x_{2n+1}$)., as shown in Equation (Equation 4) [24].
(4)$$\begin{array}{c} \begin{array}{cc} X_{k} & =\sum_{n=0}^{N-1}x_{n}e^{-\frac{2\pi kn}{N}}\\ \; & =\sum_{n=0}^{\frac{N}{2}-1}x_{2n}e^{-\frac{j2\pi k(2n)}{N}}+\sum_{n=0}^{\frac{N}{2}-1}x_{2n+1}e^{-\frac{j2\pi k(2n+1)}{N}}\\ \; & =\sum_{n=0}^{\frac{N}{2}-1}x_{2n}e^{-\frac{j2\pi kn}{N/2}}+e^{-\frac{j2\pi k}{N}}\sum_{n=0}^{\frac{N}{2}-1}x_{2n+1}e^{-\frac{j2\pi kn}{N/2}} \end{array} \end{array}$$

In this computation, the N-point DFT is conceptually divided into two halves. The first half of the DFT corresponds to samples k=0 to $k=\frac{N}{2}-1$ and is computed as the addition of two DFTs, each of $\frac{N}{2}$ points, combined with complex exponentials, as shown in Equation (Equation 4) [24]. The second half of the DFT corresponds to samples $k=\frac{N}{2}$ to k=N−1 and is also computed as the addition of the same two $\frac{N}{2}$-point DFTs, but combined with different complex exponentials. This is shown in Equation (Equation 5), where $k=\frac{N}{2}+t,0\leq t<\frac{N}{2}$ [24].
(5)$$\begin{array}{c} \begin{array}{cc} X_{k}=X_{N/2+t} & =\sum_{n=0}^{\frac{N}{2}-1}x_{2n}e^{-\frac{j2\pi (N/2+t)n}{N/2}}+e^{-\frac{j2\pi (N/2+t)}{N}}\sum_{n=0}^{\frac{N}{2}-1}x_{2n+1}e^{-\frac{j2\pi (N/2+t)n}{N/2}}\\ \; & =\sum_{n=0}^{\frac{N}{2}-1}x_{2n}e^{-\frac{j2\pi tn}{N/2}}-e^{-\frac{j2\pi t}{N}}\sum_{n=0}^{\frac{N}{2}-1}x_{2n+1}e^{-\frac{j2\pi tn}{N/2}} \end{array} \end{array}$$

In a generalization of this approach, the computation of an *N*-point DFT involves combining *P* DFTs, each of $\frac{N}{P}$ points, as illustrated in Equation (Equation 6).
(6)$$\begin{array}{c} \begin{array}{cc} X_{k} & =\sum_{p=0}^{P-1}\left(e^{-\frac{j2\pi kp}{N}}\sum_{n=0}^{\frac{N}{P}-1}x_{\left(Pn+p\right)}e^{-\frac{j2\pi kn}{\left(N/P\right)}}\right) \end{array} \end{array}$$

This method conceptualizes the computation of the *N*-point DFT in *P* sections: from samples k=0 to $k=\frac{N}{P}-1$, then from samples $k=\frac{N}{P}$ to $k=2\frac{N}{P}-1$, and so forth, up to the section comprising samples $k=\left(P-1\right)\frac{N}{P}$ to k=N−1. Each section is computed as the summation of *P* identical DFTs of $\frac{N}{P}$ points, combined with different complex exponentials. This is shown by the fact that if $k=s\frac{N}{P}+t$, where 0≤t<N/P and 0≤s<P (the *k*-th sample is in the *s*-th section), then Equation (Equation 6) can be expressed as Equation (Equation 7).
(7)$$\begin{array}{c} \begin{array}{cc} X_{k} & =X_{sN/P+t}=\sum_{p=0}^{P-1}\left(e^{-\frac{j2\pi (\frac{N}{P}s+t)p}{N}}\sum_{n=0}^{\frac{N}{P}-1}x_{\left(Pn+p\right)}e^{-\frac{j2\pi (s\frac{N}{P}+t)n}{\left(N/P\right)}}\right)\\ \; & =\sum_{p=0}^{P-1}\left(e^{-\frac{j2\pi (\frac{N}{P}s+t)p}{N}}\sum_{n=0}^{\frac{N}{P}-1}x_{\left(Pn+p\right)}e^{-\frac{j2\pi tn}{N/P}}\right) \end{array} \end{array}$$

The proposed FPGA acquisition hardware accelerator uses this approach to compute large FFTs. The FPGA computes an *N*-point FFT by adding the results of *P* FFTs, each of $\frac{N}{P}$ points, combined with different complex exponentials. This is shown in Figure 5. This method is subject to the restriction that $\frac{N}{P}$ must be a power of two. Each FFT of $\frac{N}{P}$ points is applied to one subset of the input samples $x_{\left(Pn\right)}$, $x_{\left(Pn+1\right)}$, $x_{\left(Pn+2\right)}$, …, $x_{\left(Pn+P-1\right)}$ according to Equation (Equation 6). In Figure 5, $X_{k^{\prime }}^{1}$ is the output of $FFT_{\frac{N}{P}}\left(x_{Pn}\right)$, $X_{k^{\prime }}^{2}$ is the output of $FFT_{\frac{N}{P}}\left(x_{Pn+1}\right)$, etc., and $X_{k^{\prime }}^{P}$ is the output of $FFT_{\frac{N}{P}}\left(x_{Pn+P-1}\right)$, where $0\leq k^{\prime }<\frac{N}{P}$.

A state machine in the FPGA combines each set of samples produced by the $\frac{N}{P}$-point FFTs $\left\{X_{k^{\prime }}^{1},X_{k^{\prime }}^{2},\ldots ,X_{k^{\prime }}^{P}\right\}$, with complex exponentials, adding the results to produce *P* output samples of the *N*-point FFT. More specifically, the state machine utilizes input samples $\left\{X_{k^{\prime }}^{1},X_{k^{\prime }}^{2},\ldots ,X_{k^{\prime }}^{P}\right\}$ to compute the *N*-point FFT output samples $\left\{X_{k^{\prime }},X_{\frac{N}{P}+k^{\prime }},\ldots ,X_{\left(P-1\right)\frac{N}{P}+k^{\prime }}\right\}$). A sample register is used to hold the values $\left\{X_{k^{\prime }}^{i}\right\}$, 0≤i<P, while the FPGA is computing the long FFT output samples $\left\{X_{k^{\prime }},X_{\frac{N}{P}+k^{\prime }},X_{2\frac{N}{P}+k^{\prime }},\ldots ,X_{\left(P-1\right)\frac{N}{P}+k^{\prime }}\right\}$. Because of the method employed, the *N*-point FFT output samples $X_{k}$ are computed out-of-order.

The number of FFTs (*P*), and the length of each FFT (N/P) are determined as follows:The sampling frequency is chosen so that the number of samples *N* representing 100 ms is a multiple, *P*, of a power of two. N/P is constrained to be a power of two to ensure compatibility with FFT IP cores provided by FPGA manufacturers.To minimize the computation overhead in the proposed implementation, N/P is set to its maximum feasible value. This value is determined by either the maximum transform length supported by the FFT IP cores from FPGA manufacturers or a value that ensures the sampling frequency is as close as possible to the desired value.

The details regarding the selection of the sampling frequency, the number of FFTs (*P*), and the length of each FFT (N/P) for the receiver presented in this paper are elaborated further in Section 3.4.3.

#### 3.4.3. Implementation

The acquisition process, when operating in high-sensitivity mode, incorporates both the PCPS algorithm [23] and the GPDIT PDI strategy [7]. The acquisition algorithm computes the CAF using a CI time of 100 ms. The CAF is computed as the circular correlation between the received signal and a local replica of the Galileo pilot codes, using various trial Doppler frequencies. The circular nature of the correlation is ensured by the fact that the Galileo tiered codes have a duration of 100 ms. The outcome of up to 7 consecutive CAFs are merged using the GPDIT technique shown in Equation (Equation 3), resulting in a maximum total integration time of 700 ms. The acquisition uses a Constant False Alarm Rate (CFAR) detector. Table 4 shows the input parameters of the acquisition algorithm.

Table 5 shows the output parameters of the acquisition algorithm for cases where a positive acquisition is obtained.

Table 6 shows the variables introduced in the description of the high-sensitivity algorithm.

The proposed acquisition process is detailed in Algorithm 1. In the initial step, the acquisition buffers 700 ms of samples, equivalent to $N\;\cdot \;N_{nc}$ samples, where *N* represents the number of samples used for CI (equivalent to 100 ms), and $N_{nc}$ is 7, representing the maximum number of GPDIT non-coherent combinations used for detecting weak signals. For each GPDIT iteration, the algorithm takes 100 ms worth of samples (referred to as x[n]) from the acquisition buffer $x_{IN}\left[n\right]$ and performs a Doppler frequency search on x[n], from the minimum Doppler frequency $f_{min}$ to the maximum Doppler frequency $f_{max}$ in $f_{step}$ steps. For each tested Doppler frequency $f_{d}$, the acquisition conducts Doppler wipeoff and performs a circular correlation between the received signal x[n] and Galileo pilot codes in the frequency domain C[k], obtaining the CAF $R_{k}\left(\tau ,f_{d}\right)$. The CAF is used to compute the corresponding GPDIT iteration $Z_{\mathrm{GPDIT}}\left(\tau ,f_{d}\right)$. The algorithm then searches the time-Doppler frequency grid for the peak value of the computed GPDIT iteration, obtaining the peak value along with an estimation of the Doppler frequency and code phase $\left\{S_{max},f_{i},\tau_{j}\right\}$. Finally, it evaluates a Generalized Likelihood Ratio Test (GLRT) function $\Gamma_{\mathrm{GLRT}}$, and determines whether the searched signal is detected. If positive, the algorithm declares successful acquisition; otherwise, it returns to the main loop to compute the next successive CAF and the next GPDIT iteration. If a total of 7 GPDIT combinations are computed and the signal is not detected, then the algorithm declares negative acquisition.
**Algorithm 1** High-sensitivity acquisition1:Set the total number of GPDIT iterations $N_{nc}=7$2:Buffer 700 ms worth of samples: $x_{IN}\left[0\right],x_{IN}\left[1\right],\ldots ,x_{IN}\left[N\;\cdot \;N_{nc}\right]$3:Set the current GPDIT iteration $n_{nc}=1$4:While $n_{nc}\leq N_{nc}$ do5: Take the next 100 ms worth of samples from the acquisition buffer: $x\left[n\right]=\{x_{IN}\left[\left(n_{nc}-1\right)\;\cdot \;N\right],x_{IN}\left[\left(n_{nc}-1\right)\;\cdot \;N+1\right],\ldots ,x_{IN}\left[\left(n_{nc}-1\right)\;\cdot \;N+N-1\right]\}$6: For $f_{d}=f_{min}$ to $f_{d}=f_{max}$ in $f_{step}$7:  Perform Doppler wipe-off: $x_{d}\left[n\right]=x\left[n\right]\;\cdot \;e^{-j2\pi f_{d}nT_{s}}$, for n=0,…,N−18:  Compute $X_{d}\left[k\right]=FFT_{N}\left(x_{d}\left[n\right]\right)$9:  Compute $Y\left[k\right]=X_{d}\left[k\right]\;\cdot \;C\left[k\right]$, for k=0,…,N−110:   Compute $R_{n_{nc}}\left(\tau ,f_{d}\right)=\frac{1}{N^{2}}IFFT_{N}\left(Y\left[k\right]\right)$11:   Compute $Z_{\mathrm{GPDIT}}\left(\tau ,f_{d}\right)=\sum_{k=1}^{n_{nc}}|R_{k}\left(\tau ,f_{d}\right){|}^{2}+2\left|\sum_{k=2}^{N_{nc}}R_{k}\left(\tau ,f_{d}\right)R_{k-1}^{*}\left(\tau ,f_{d}\right)\right|$12: End for13: Search the peak value and its indices in the search grid: $\left\{S_{max},f_{i},\tau_{j}\right\}=max_{f,\tau }Z_{\mathrm{GPDIT}}\left(\tau ,f_{d}\right)$14: Compute input signal power estimation ${\hat{P}}_{IN}=\frac{1}{N}\sum_{n=0}^{N-1}{\left|x\left[n\right]\right|}^{2}$15: Compute the GLRT function with normalized variance $\Gamma =\frac{2NS_{max}}{{\hat{P}}_{IN}}$16: If $\Gamma_{\mathrm{GLRT}}>\gamma$ (Compare with threshold value)17:  Declare positive acquisition and provide $f_{d_{acq}}=f_{i}$ and $\tau_{acq}=\tau_{j}$18:  Break;19: Else20:  If $n_{nc}=N_{nc}-1$ (last GPDIT iteration)21:   Declare negative acquisition22:  End if23: End if24:End while

The acquisition algorithm is executed collaboratively by the FPGA and the embedded processor, with each component handling a distinct portion of the process. The FPGA is responsible for capturing the samples from the analog front-end and computing the successive CAFs using the captured samples. Concurrently, the embedded processor executes the GPDIT iterations and the CFAR detector. Even though the FPGA manages the majority of the computational workload related to the acquisition process, the embedded processor is actively involved in the acquisition tasks.

The embedded processor can simultaneously handle the acquisition process and the other tasks required for the baseband software processing engine. These tasks include the control of the FPGA tracking multicorrelator hardware accelerators, the execution of the telemetry decoders, the computation of GNSS basic measurements, and the derivation of the navigation solutions. The Zynq UltraScale+ XCZU9EG-2FFVB1156 All-Programmable MPSoC [28] used to demonstrate the proposed design, features a quad-core embedded processor. Therefore, even if one processor core is devoted to the acquisition process, the other three processor cores remain available for the other tasks, enabling the real-time processing of GNSS signals.

Figure 6 shows a detailed view of the FPGA acquisition hardware accelerator depicted in Figure 3. The PL DDR4 memory component shown in Figure 6 is the same as the one shown in Figure 3. The acquisition hardware accelerator comprises two modules: the Sample Capture module and the CAF computation module. When performing acquisition in high-sensitivity mode, the Sample Capture module collects 700 ms worth of samples from the analog front-end and stores the samples into the PL DDR4 memory component. Once the first 100 ms worth of samples has been captured, concurrently with the remaining part of the sample capture process, the CAF computation module initiates the calculation of the successive CAFs. By overlapping the CAF computation and the sample capture, the acquisition latency is reduced. The FPGA writes the computed CAFs to the PL memory. Concurrently, the embedded processor reads the computed CAFs from the PL memory component and executes the GPDIT iterations. The embedded processor executes the GPDIT iterations simultaneously to the FPGA calculating the CAFs, to further reduce the acquisition latency. In accordance with the architecture presented in [21], the high-sensitivity acquisition hardware accelerator exposes memory-mapped registers to the embedded processor for operation control. Additionally, it is equipped to issue interrupt requests to the embedded processor, facilitating the efficient synchronization of operations.

Figure 7 presents a timing diagram for the acquisition process, illustrating the concurrent computation of successive CAFs ($R_{k}\left(\tau ,f_{d}\right)$) and GPDIT iterations ($Z_{{\mathrm{GPDIT}}_{k}}\left(\tau ,f_{d}\right)$) within the FPGA and the embedded processor, respectively. In the figure, time progresses from left to right. The FPGA and the embedded processor synchronize their operations by means of interrupt requests (marked as “Int.” in Figure 7), going from the FPGA to the PS. This process is explained below in more detail.

The embedded processor commences the acquisition process, as indicated by ‘Initiate Acquisition Process’ in Figure 7, which includes directing the FPGA to capture samples, a step referred to as ‘Initiate Sample Capture’ in the same figure. This involves the FPGA collecting 700 ms worth of samples in 100 ms increments. After capturing the first segment, the FPGA generates an interrupt request to the embedded processor. The processor, in response, instructs the FPGA to start the computation of the first CAF at the initial tested Doppler frequency ($R_{k}\left(\tau ,f_{min}\right)$). This instruction from the processor to the FPGA is denoted as ‘Initiate CAF Computation’ in Figure 7. Following this, the FPGA and embedded processor work in sequence to compute the CAFs and GPDIT iterations for each Doppler frequency.

For each frequency, the FPGA performs circular cross-correlation between the received signal and the local replica. After completing this computation, it issues an interrupt request to the embedded processor. Upon receiving the interrupt, the processor initiates the corresponding GPDIT iteration for that Doppler frequency. Simultaneously, it commands the FPGA to calculate the CAF for the next frequency. This command is not shown in Figure 7 to maintain clarity. This procedure is systematically repeated across all Doppler frequencies, progressing from $f_{d}=f_{min}$ to $f_{d}=f_{max}$ in increments defined by $f_{step}$. Through this orchestrated approach, the FPGA consistently stays ahead by computing the CAF for the next frequency while the embedded processor concurrently handles the GPDIT iteration for the current frequency. Specifically, the FPGA computes the CAF for the (n+1)-th Doppler frequency ($R_{k}\left(\tau ,f_{min}+n\;\cdot \;f_{step}\right)$), while the embedded processor executes the GPDIT iteration for the *n*-th frequency ($Z_{\mathrm{GPDIT}k}\left(\tau ,f_{min}+\left(n-1\right)\;\cdot \;f_{step}\right)$).

With this approach, upon completion of the computation of the $n_{nc}$-th CAF, the FPGA initiates the computation of the subsequent ($n_{nc}$+1)-th CAF in parallel with the embedded processor finishing the execution of the $n_{nc}$-th GPDIT iteration. Importantly, this parallel computation is speculative, meaning the results of the ($n_{nc}$+1)-th CAF might not be utilized if the CFAR detector signals a positive detection at the conclusion of the $n_{nc}$-th GPDIT iteration. Adopting this strategy minimizes acquisition latency through the simultaneous computation of CAF and GPDIT iterations.

Figure 8 shows a detailed block diagram of the CAF computation module shown in Figure 6, as it is implemented in the FPGA. The CAF computation module performs Doppler wipeoff and computes the circular correlation between the received signals and the local replica of the pilot tiered code in the frequency domain. It has four memory interfaces implemented using the AMBA AXI4-Lite protocol specification [34]: AXI4-Lite interface 1 fetches the FFT of the local replica of the pilot-tiered code from the PL memory component. AXI4-Lite interface 2 fetches the captured samples from the analog front-end, which are also stored in the PL DDR4 memory component. AXI4-Lite interface 3 stores the computed CAF in the PL DDR4 memory. These three memory interfaces perform read and write transactions to and from the PL DDR4 simultaneously, thus decreasing the acquisition latency. AXI4-Lite interface 4 exposes a bank of memory-mapped registers to the embedded processor. The embedded processor uses these registers to configure and control the acquisition process. Furthermore, the CAF computation module also incorporates the following units, as depicted in Figure 8: a Doppler wipeoff unit, a unit for computing large FFTs and IFFTs (the large FFT/IFFT block), a unit for multiplying the FFT output with the local replica of the pilot tiered code (the Enable/Disable Local Code Mult block), and a unit for reporting a scaling factor to the embedded processor. This scaling factor is automatically applied to the FFT/IFFT output to prevent saturation, and it is referred to as the block exponent in Figure 8. Finally, a state machine controls the execution of the CAF computation, which occurs in two steps:1.In the first step, the CAF computation module computes the frequency domain representation of the circular correlation: the module fetches 100 ms of samples from the PL memory component, performs Doppler wipeoff, computes the FFT of the Doppler-corrected signal, and multiplies the results by the FFT of the local replica of the pilot tiered code. Finally, the result of this multiplication is stored back into the PL memory component. Storing intermediate results in the PL memory component reduces FPGA memory usage. During this first step, the Enable/Disable Doppler wipeoff block and the Enable/Disable FFT Multiplication block shown in Figure 8 are enabled. The local carrier used for Doppler wipeoff is efficiently implemented using the Coordinate Rotation Digital Computer (CORDIC) algorithm [38].2.In the second step, the CAF computation module calculates the time-domain representation of the circular cross-correlation. It retrieves the results of the multiplication of the FFT by the local replica obtained in step 1, performs the IFFT, and then stores the results back in the PL memory component.

The large FFT/IFFT module depicted in Figure 8 computes the FFT and the IFFT, applying a scaling factor to the input signals to prevent overflow in the calculations. The scaling factor applied is made available to the embedded processor through the Report Block Exponent block. The embedded processor normalizes the application of the scaling factor to ensure uniform scaling across all Doppler frequencies and GPDIT iterations.

Figure 9 shows a block diagram of the large FFT/IFFT module depicted in Figure 8. This module computes large FFTs and IFFTs by combining multiple small FFTs and IFFTs with complex exponentials, extending the transform size. This approach is explained in Section 3.4.2 and illustrated in Equation (Equation 6), where an *N*-point DFT is computed as a weighted combination of *P* DFTs, each of length N/P. The same principle applies for the computation of the IFFT.

The 64k-FFT/IFFT module, shown in Figure 9, is implemented using AMD’s FFT LogiCORE IP v9.1 [39]. This FFT core imposes a maximum transform size limit, restricting it to $2^{16}$ = 64k samples. To accommodate this limitation, the receiver’s sampling frequency is deliberately chosen to ensure that the computation of a 100 ms circular cross-correlation in the frequency domain results in an FFT size that is a multiple of 64k samples. Consistent with this, the sampling frequency is set to 15.728640 Mega samples per second (Msps), which provides sufficient capability to track GNSS signals in both the L1/E1 and L5/E5a bands [40]. As shown in Figure 3, the acquisition hardware accelerator incorporates a downsampling filter in the L1/E1 band, reducing the sampling frequency by a factor of 4 to 3.932160 Msps. Consequently, to compute the CAF using a CI time of 100 ms, FFTs/IFFTs with a transform size of 393,216 samples are required. This transform size is equivalent to 6×
64k samples. To efficiently handle this calculation, the FPGA computes FFTs of 393,216 samples by combining six FFTs, each of 64k samples, following the approach depicted in Figure 5.

The LogiCORE FFT is configured to implement a 6-channel 64k-FFT on the incoming samples. An input sample selector cyclically distributes the incoming samples to the FFT channels in a sequential order (1st sample to the 1st channel, 2nd sample to the 2nd channel, and so forth. In the complex exponential combining block, a state machine combines the outputs of the 64k-FFT with the complex exponentials $e^{-j2\pi \frac{k}{N}s}$, as illustrated in Equation (Equation 6). The complex exponentials are implemented using the CORDIC algorithm. The LogiCORE FFT automatically applies a scaling factor (block exponent) to each channel to prevent overflow. The Apply Scaling Factors module in Figure 9 dynamically equalizes the various FFT channels by ensuring that the same scaling factor is uniformly applied to all of them. The scaling factor used is reported to the embedded processor. Finally, the 64k-FFTs, combined with the complex exponentials, are added to obtain the 393,216-point FFT results. The 393,216-point FFT and IFFT output samples are produced out of order. However, these samples are automatically reordered during the storage process to memory, minimizing latency.

### 3.5. Doppler Prediction

The high-sensitivity acquisition parallelizes the code phase search by performing a circular cross-correlation between the received GNSS signal and a local replica of the signal’s tiered code. However, this correlation has to be performed once for each possible trial Doppler frequency [23]. In cases where a static GNSS receiver lacks prior knowledge of the received Doppler frequencies, it is obligated to conduct a search spanning from −5 kHz to +5 kHz. As shown in Section 3.4.1, when operating in high-sensitivity mode, the use of a CI time of 100 ms yields a Doppler search step of 5 Hz. Consequently, the acquisition process involves testing 2001 Doppler frequencies, conducting a frequency sweep across all possible frequencies within ±5 kHz of the nominal carrier frequency in increments of 5 Hz. This leads to a significant acquisition latency.

When the receiver is processing signals in real time, the acquisition latency cannot be arbitrarily large. The key factors limiting the maximum acceptable latency are the maximum allowable TTFF and the degradation over time of the parameters estimated during acquisition, such as timing synchronization and Doppler frequency. This degradation is caused by the receiver clock drift and the continuously changing Doppler frequency in the received signals. If the acquisition process is slow, by the time the receiver initiates tracking the detected signal, the Doppler frequency has already changed. This, coupled with local crystal oscillator inaccuracies, diminishes the accuracy of the Doppler frequency and code phase estimated during the acquisition process.

The receiver employs the assistance data to predict the Doppler frequency of the incoming signals, leading to a reduction in the Doppler search space during acquisition. The Doppler search is performed considering potential inaccuracies in the Doppler prediction, and the presence of a Carrier Frequency Offset (CFO) in the analog front-end, originating from deviations in the accuracy of the local crystal oscillator.

Despite the use of Doppler prediction, the latency of the high-sensitivity acquisition process may still remain large, decreasing the probability of successfully tracking the detected signals. To address this problem, the receiver offers an option to execute the acquisition process in two stages. The first stage performs a relatively large Doppler search around the predicted Doppler frequency. Then, in the second stage, the acquisition process is repeated, capturing new samples from the analog front-end and employing a very small Doppler search space centered on the Doppler frequency identified in the first stage. The execution of stage 2 is much faster than the execution of stage 1 because of the smaller Doppler search space. While this two-stage execution increases overall acquisition latency and may influence the probability of detection, stage 2 reduces the latency from sample capture to the initiation of the tracking process, thereby enhancing the accuracy of the parameters estimated during acquisition. The two-step acquisition process is shown in Table 7.

The method outlined in Table 7 is effective provided that, during the execution of stage 1, the change in the Doppler frequency of the received signal remains within the Doppler search range specified for stage 2. Otherwise, the signal detected in stage 1 may not be detected in stage 2. The probability of detection may also be affected by the need to detect the same signal twice.

### 3.6. Acquisition in Normal-Sensitivity Mode

When working in normal-sensitivity mode, the acquisition process detects GPS L1 C/A, GPS L5, Galileo E1b/c, and Galileo E5a signals received at nominal $C/N_{0}$ levels. Additionally, the acquisition process can be configured to detect either the data component or the pilot component of the received signals. Much like the procedure in high-sensitivity mode, the acquisition process involves conducting a cross-correlation between the received signal and a local replica of the satellite PRN code. However, this correlation is performed using a short CI time, equal to the length of the PRN code (1 ms for GPS L1 C/A, GPS L5, and Galileo E5a; 4 ms for Galileo E1b+c) or a small multiple of this length. The results from several consecutive CAFs $R_{k}\left(\tau ,f_{D}\right)$ can be effectively combined using the non-coherent Post Detection Integration (NPDI) technique, shown in Equation (Equation 8) [7], where $Z_{\mathrm{NPDI}}$ is the result of the NCI. Throughout this process, a CFAR detector is employed.
(8)$$Z_{\mathrm{NPDI}}=\sum_{k=1}^{N_{nc}}{\left|R_{k}\left(\tau ,f_{D}\right)\right|}^{2}$$

When operating in normal-sensitivity mode, the high-sensitivity acquisition hardware accelerator is responsible for capturing samples from the analog front-end. Subsequently, the embedded processor takes charge of all calculations, including Doppler wipe-off, the circular cross-correlation, and the computation and the non-linear combinations of consecutive CAFs. The rationale behind this arrangement lies in the current design of the high-sensitivity acquisition hardware accelerator in the FPGA. Specifically tailored for acquiring weak signals using long CI times, the current version lacks the flexibility required to compute the CAF in normal-sensitivity mode. Nonetheless, the Ultrascale+ XCZU9EG MPSoC’s embedded processor boasts ample computing power to execute the acquisition in normal-sensitivity mode in real time, manage the FPGA tracking multicorrelators, run the telemetry decoders, compute the observables, and determine the PVT.

### 3.7. Tracking

The role of a tracking algorithm is to follow the evolution of the signal synchronization parameters: code phase τ(t), Doppler frequency $f_{d}\left(t\right)$ and carrier phase ϕ(t). The proposed receiver implements tracking loops to continuously monitor and adjust to the code and carrier parameters of the incoming signal. Specifically, a Delay Lock Loop (DLL) is employed to track the signal’s code delay, a Phase Lock Loop (PLL) is dedicated to monitoring and adjusting to the signal’s phase, and a Frequency Locked Loop (FLL) can be enabled to monitor the signal’s frequency.

The tracking multicorrelator hardware accelerators in the FPGA perform the Doppler wipe-off and the multicorrelation of the incoming signal with the local replica of the PRN codes. The embedded processor executes the PLL, the FLL, and the DLL. The PLL and the FLL employ a four-quadrant arctangent discriminator. The DLL employs a noncoherent Very Early Minus Late Power (VEMLP) normalized discriminator when tracking Galileo E1b/c signals, and a noncoherent Early Minus Late envelope-normalized discriminator when tracking Galileo E5a, GPS L1 C/A, and GPS L5 signals.

The tracking multicorrelator hardware accelerators are configured as shown in Table 8. When tracking GPS L1 C/A signals, the receiver follows the data component of the received signals. For other signals, the receiver tracks the pilot component, utilizing an additional correlator dedicated to demodulating the data component.

For details on the implementation of the tracking multicorrelator hardware accelerators, refer to [21].

#### 3.7.1. Tracking in High-Sensitivity Mode

When tracking Galileo E1b/c signals in high-sensitivity mode, the receiver initiates signal tracking with synchronization to the pilot’s secondary code. This synchronization is achieved through the acquisition process, which computes the CAF using a circular cross-correlation with the complete Galileo pilot tiered code, including both the primary and secondary codes. Consequently, the tracking process can commence with an extended CI time that surpasses the duration of the PRN code. The receiver is configured to track the pilot signals, which have predictable tiered codes. This configuration enables the utilization of CI times that exceed the duration of a single data bit in the tracking process. These capabilities are essential for tracking weak signals, addressing the challenges posed by potential bit errors in the demodulated data due to low C/N0 conditions [36,37]. To effectively track weak signals, it is necessary to employ an extended CI time. However, compared to signal acquisition, a shorter CI time can be utilized for the tracking process, due to the fundamental differences between tracking and acquisition within signal processing. Tracking is an estimation problem that benefits from prior knowledge about the code phase and the carrier phase. This information not only enhances the sensitivity of the tracking loops but also allows for refined estimations of these parameters. In contrast, acquisition is primarily a detection problem, focused on initially identifying the signal’s presence without the benefit of prior accurate information. In line with this, the receiver was tested using a tracking CI time of 40 ms, which is smaller than the CI time used in acquisition (100 ms). For more details on the configuration of the tracking loops, refer to Appendix A.

The current implementation of the tracking algorithm does not include any type of multipath mitigation. The high sensitivity tracking mode is mainly designed to track weak GNSS signals.

#### 3.7.2. Tracking in Normal-Sensitivity Mode

In normal-sensitivity mode, the receiver initiates tracking of detected signals without synchronization to the pilot’s secondary code or the telemetry preambles. The tracking process commences with a CI time equivalent to the duration of the PRN primary code. Once synchronization with the pilot’s secondary code or the telemetry preambles is achieved, the tracking process dynamically increases the CI time based on user configuration.

### 3.8. Telemetry Decoding

When tracking very weak Galileo E1b/c signals, the receiver may not be able to correctly demodulate the telemetry messages due to the presence of bit errors [36,37]. However, demodulating the received telemetry messages is needed to obtain the TOW of the received messages. The TOW is used to compute the GNSS basic measurements and the navigation solutions. For this reason, the GNSS-SDR software telemetry decoder was upgraded to estimate the TOW of the received Galileo E1b/c messages using the Time of Arrival (TOA) of the telemetry synchronization patterns and the receiver’s local clock. The receiver performs TOW estimation in three steps: sync frame detection, sync frame confirmation, and TOW estimation. These steps are explained below.

#### 3.8.1. Sync Frame Detection

The Galileo telemetry decoder initiates its process by identifying the I/NAV synchronization patterns within the received telemetry frames. To address the potential presence of bit errors, the receiver maintains a list of sample stamps corresponding to the most recently detected synchronization patterns. These sample stamps represent the receiver sample counter associated with those patterns. For each identified synchronization pattern, the receiver verifies if the timing relative to any of the most recently detected patterns is a multiple of the Galileo E1 I/NAV synchronization pattern period. If this condition is met, then the telemetry decoder assumes sync frame detection and proceeds to the second step, which involves sync frame confirmation.

#### 3.8.2. Sync Frame Confirmation

The telemetry decoder checks that at least a certain percentage of preambles is accurately detected at the expected timings. If the detection of the telemetry sync patterns is confirmed, then the receiver assumes correct sync frame detection and proceeds to the third step, which involves TOW estimation. Otherwise, the receiver assumes a synchronization issue and goes back to the sync frame detection.

#### 3.8.3. TOW Estimation

The receiver estimates the TOW using the TOA of the telemetry synchronization patterns and the local embedded OS clock. The TOW of the Galileo synchronization patterns corresponds to the exact timing of the telemetry page frame boundaries [41]. The receiver takes advantage of the fact that, in the Galileo E1b/c signals, the synchronization patterns are transmitted in sync with the second. The estimation of the TOW involves measuring the exact TOA of the telemetry sync patterns and rounding this time to the nearest second. The TOA is computed in Galileo System Time (GST) as follows: In the first step, the local time is converted to UTC considering the local time zone. In the second step, the computed UTC time is converted to GST considering the leap seconds between UTC and GST. Assuming that the GNSS signal travel time, plus the error induced by delays in the receiver detecting the preambles, added to the receiver’s local clock offset is in total below ±500 ms, the computed TOW estimation is deemed correct. This is because rounding the TOA to the nearest second provides the TOW. Achieving the required accuracy in the local OS clock is performed by synchronizing the receiver’s OS clock to the true local time using the Network Time Protocol (NTP) [42]. The NTP protocol offers a nominal accuracy of tens of milliseconds on Wide Area Networks (WANs) [42], which is sufficient for the proposed TOW estimation process. When in the TOW estimation state, the telemetry decoder continuously verifies that a minimum percentage of preambles is accurately detected at the expected timings. If the receiver fails to detect the telemetry preambles, it assumes a synchronization issue and reverts to the frame sync detection state. This allows the receiver to resynchronize with the received preambles in case of an incorrect sync frame detection.

When working in post-processing mode using recorded GNSS signals, the receiver local OS clock time does not correspond to the recorded signal time. Therefore, the receiver embedded OS clock cannot be used to estimate the TOA of the telemetry sync patterns. In this case, the telemetry decoder estimates the TOW of the telemetry messages using the receiver sample counter and a predetermined relationship between the sample counter and the recorded signal time that shall be provided by the user. The sample counter counts up the number of processed samples and is automatically reset when the user starts the GNSS-SDR software receiver.

### 3.9. Computation of the Navigation Solutions

The embedded processor computes navigation solutions using the GNSS basic measurements: pseudorange, carrier phase, and Doppler shift. The computation of the navigation solutions is implemented using the RTKLIB open-source package for standard and precise positioning with GNSS [43].

The estimated receiver positions might include invalid solutions due to unmodeled measurement errors. The processor performs a series of validation tests on the estimated receiver positions, including a residuals test and a Geometric Dilution of Precision (GDOP) Test [43]. If any of these tests fails, the solution is rejected as an outlier. In addition, the software implements Receiver Autonomous Integrity Monitoring Fault Detection and Exclusion (RAIM-FDE) [44]. RAIM-FDE attempts to exclude invalid measurements due to satellite malfunction, receiver fault, or large multipath by iteratively estimating the receiver’s position while excluding one visible satellite at a time.

When operating in high-sensitivity mode, the receiver utilizes assistance ephemeris data to determine PVT. This is necessary because the receiver may encounter challenges in correctly demodulating the telemetry messages of weak Galileo E1b/c signals, primarily due to the presence of bit errors [36,37]. If the telemetry decoder successfully demodulates the telemetry messages during operation, the receiver then updates the assistance ephemeris data with the latest information received from the satellites. Assistance ephemeris data can be optionally employed for the GPS and the Galileo E5a signals as well.

### 3.10. GNSS Output Products

The proposed receiver produces GNSS signal products in standard open formats, enabling integration with other positioning technologies where available. These formats include Receiver-Independent Exchange Format (RINEX) Files, Radio Technical Commission for Maritime Services (RTCM) messages with configurable rates, GPS Exchange Format (GPX), Keyhole Markup Language (KML), GeoJSON, and National Marine Electronics Association (NMEA-0183) messages for sensor integration. PVT solutions are generated at a configurable rate in the World Geodetic System (WGS-84) based on Least Squares or Kalman filtering. For each visible satellite, the receiver provides time-tagged measurements of pseudorange (in m), carrier phase (in cycles) or phase range (in m), Doppler shift (in Hz) or pseudorange rate (in m/s), received signal strength (in dB-Hz), Dilution of Precision (DOP), raw navigation data, tracking correlators’ output, position and Velocity (WGS-84), GPS Time, Galileo Time, and UTC time.

## 4. Results

We assessed the performance of the proposed receiver based on several criteria. These included acquisition latency, the receiver’s ability to process Galileo E1b/c signals in high-sensitivity mode, and its capability to simultaneously process Galileo E1b/c signals in high-sensitivity mode alongside Galileo E5a and GPS signals in normal-sensitivity mode. The precision of navigation solutions and the power consumption were also evaluated.

The receiver’s ability to process Galileo E1b/c signals in high-sensitivity mode was tested using three different TCXOs and an OCXO. Two TCXOs caused a CFO in the received signals larger than expected, requiring the user to perform a manual estimation and correction of the CFO before using the receiver in high-sensitivity mode. This procedure was not required when using an OCXO, with a better frequency stability. A detailed explanation is provided in Section 4.2. For this reason, the authors switched to using an OCXO for the remaining tests. However, improving the acquisition algorithm to eliminate the need for manual CFO compensation remains as future work.

Section A.1 provides the receiver configuration used for processing the Galileo E1b/c signals in high-sensitivity mode, while Section A.2 offers the configuration for processing the Galileo E5a and GPS signals in normal-sensitivity mode. Section A.3 details the receiver’s PVT configuration.

### 4.1. Test Setup

Figure 10 illustrates the block diagram of the test setup used to apply live signals to the proposed receiver. The antenna captures GNSS signals from the sky. A variable attenuator is manually adjusted to decrease the carrier-to-noise density ratio ($C/N_{0}$) of the received signals down to 20 dB-Hz. Achieving a uniform $C/N_{0}$ of 20 dB-Hz for all signals proves challenging due to variations in the received signal power from different satellites. Splitter 1 divides the received signals into two paths: one leading to the device under test and the other to a commercial receiver, which is used to validate the $C/N_{0}$ of the received signals. The receiver introduced in this paper uses splitter 2 to divide the incoming signal internally, distributing it between the two RF inputs of the AD-FMCOMMS5-EBZ analog front-end. RF1 is tuned to the E1/L1 frequency band, while RF2 is tuned to the E5a/L5 band.

The test setup involves two splitters in the signal path of the high-sensitivity GNSS receiver (splitter 1 and splitter 2). Similarly, the signal path of the commercial receiver incorporates splitter 1 and splitter 3. Each splitter introduces a 3-dB attenuation in the received signal. This arrangement allows us to maintain equal signal conditions, as detailed in the following explanation.

The $C/N_{0}$ observed by both receivers can be computed using the Friis formulas for noise [2]. Without the variable attenuator, the $C/N_{0}$ seen by the receivers is mainly influenced by factors like received signal strength, antenna noise temperature, and the noise figure of the Low Noise Amplifier (LNA) in the active antenna. The splitters in the signal path, including splitter 1, splitter 2, and splitter 3, have negligible effects on the $C/N_{0}$. However, when the variable attenuator introduces substantial attenuation, the $C/N_{0}$ observed by the receivers is primarily determined by the total attenuation from the antenna to the analog front-end. This total attenuation encompasses both the variable attenuator and the attenuation introduced by the splitters. Splitter 3 introduces an attenuation of 3 dB, ensuring that the commercial receiver perceives the same $C/N_{0}$ as the receiver under test when the variable attenuator introduces significant attenuation.

The authors validated the $C/N_{0}$ of received Galileo E1b/c signals using a U-blox NEO-M8T receiver [45]. The variable attenuator was adjusted to decrease the carrier-to-noise density ratio ($C/N_{0}$) of the received signals. This adjustment aimed to reduce the $C/N_{0}$ of the weakest Galileo E1b/c signals to as low as 20 dB-Hz, while maintaining consistent attenuation levels across other signals.

The rooftop GNSS antenna facility of the GESTALT testbed [46] was utilized to enhance satellite visibility during the tests. Additionally, the tests evaluating the receiver’s ability to process Galileo E1b/c, Galileo E5a, GPS L1 C/A, and GPS L5 signals (Section 4.3) were conducted using both the rooftop GNSS antenna facility and the receiver’s TW8825 active antenna.

### 4.2. High-Sensitivity Acquisition Latency

The authors assessed the latency of the two-stage acquisition scheme by conducting evaluations with both a TCXO and an OCXO, using the test setup described in Section 4.1. The variable attenuator was used to decrease the $C/N_{0}$ of the received signals to as low as 20 dB-Hz. The GESTALT testbed rooftop GNSS antenna facility [46] was used for the tests. The acquisition latency was assessed using the procedure shown in Table 9. The software was temporarily modified to measure the worst-case acquisition latency as follows: the acquisition process consistently performed seven GPDIT iterations, with the receiver recording the acquisition latency during each execution. The latency was measured as the time it takes for the receiver to execute each acquisition stage.

#### 4.2.1. Tests Results Using the TCXO

When using the TCXO, a significant CFO was observed in the received signals. To determine the required Doppler search space for acquisition, the authors used the receiver to perform a coarse estimation of the CFO. To perform this estimation, the receiver operated in normal-sensitivity mode to predict the Doppler frequency of the tracked satellites. It then compared this prediction to the measured Doppler frequency, and averaged the error. Three different instances of TCXOs were tested. Each TCXO was tested multiple times over the course of a week. Table 10 shows the minimum and maximum estimated CFO values in the E1 frequency band for each TCXO over that week.

The measurements shown in Table 10 indicate a frequency stability of approx. 56 ppb, 99 ppb, and 170 ppb for TCXO 1, TCXO 2, and TCXO 3, respectively. A certain variation in the detected CFO is expected, given that the stability of the TCXO can be influenced by environmental factors including temperature, voltage and load variations, as well as long term frequency drift [47]. However, the CFOs induced by TCXOs 2 and 3 were higher than expected. The TCXOs were mounted on a custom-made PCB, which was attached to the RFFE and exposed to the environment. The exploration of the underlying reasons behind the observed frequency stability fluctuations in the TCXOs lies beyond the scope of this study, as this paper is focused on GNSS.

The large CFO observed in TCXO 2 and TCXO 3 required the use of a large Doppler search space in the acquisition. To mitigate the CFO, we implemented a crude two-step CFO estimation and correction process in the software. The first step required taking the receiver outdoors and using the receiver in normal-sensitivity mode. The receiver estimated the CFO during tracking using the method described above. In the second step, the receiver was tested in high-sensitivity mode, and the estimated CFO was corrected by adjusting the tuning frequency in the E1/L1 and the E5a/L5 frequency bands accordingly.

When using the TCXO, the two-stage acquisition process was set as follows: the Doppler search space for the first acquisition stage was configured to accommodate Doppler prediction inaccuracies and the CFO observed in the L1/E1 frequency band, but it was limited to ±100 Hz to limit memory usage and to prevent a large TTFF. The Doppler search space for the second stage was set to be small enough to minimize the latency from sample capture to the beginning of the tracking process, ensuring successful tracking of the detected signals. Additionally, the following constraint was considered: in the worst-case scenario, the change in the Doppler frequency of the received signal during the time it takes to go from stage 1 to stage 2 should not exceed the Doppler search space set for the stage 2. As explained in Section 3.5, failure to adhere to this constraint could result in the inability of the second stage to identify the signal detected during the first stage. In line with this, the acquisition was configured as shown in Table 11.

When utilizing a TCXO with the configuration outlined in Table 11, the acquisition process takes a maximum of 14.1 s to transition from stage 1 to stage 2 in the worst-case scenario. Considering that the worst-case Doppler shift rate experienced by a static receiver on the Earth’s surface is approximately 1 Hz/s [48], the Doppler frequency of the received signal may change by 14.1 Hz during the transition form stage 1 to stage 2. The Doppler search space used in the second stage is ±15 Hz, which is larger than 14.1 Hz. Therefore, any signal detected in the first stage can be detected in the second stage.

The limitation of the Doppler search space to ±100 Hz necessitates the use of CFO compensation and correction, due to the large CFO observed in TCXO 2 and TCXO 3. For this reason, the receiver was tested using the two-stage acquisition process configured according to Table 11, along with the CFO estimation and correction procedure. The receiver successfully acquired the weak GNSS signals listed in Table 12, initiated tracking of them, and detected their telemetry preambles. The advantage of utilizing the second acquisition stage became evident, as the receiver failed to track the detected signals if the second stage was disabled. When using the TCXOs, some channels experienced occasional loss of lock during tracking. This issue was related to the configuration of the tracking loops (see Section A.1) and the measured TCXO frequency stability shown in Table 10. This problem did not occur when using the OCXO (see Section 4.2.2).

#### 4.2.2. Tests Results Using the OCXO

We also tested the acquisition using an OCXO. The OCXO did not require any CFO correction. The OCXO was accurately tuned using a trimmer when used for the first time, and required no further adjustments. When using an OCXO, the receiver was configured according to Table 13. The Doppler search space of ±50 Hz accounts for possible inaccuracies in the estimation of the Doppler frequency and in the OCXO calibration.

As shown in Table 13, when using the OCXO, stage 1 takes 7.1 s to complete. Considering that the largest Doppler shift rate experienced by a static receiver is about 1 Hz/s [48], the Doppler shift in the received signal may vary by up to 7.1 Hz when transitioning from stage 1 to stage 2. This change is larger than acquisition Doppler search step, which is set to 5 Hz. Consequently, if the second acquisition stage is not utilized, the Doppler frequency of the received signal might not align with the Doppler frequency estimated during acquisition upon initiation of the tracking process.

To mitigate this issue, the two-stage acquisition process was employed also for the OCXO. As explained in Section 3.5, stage 2 captures new samples from the analog front-end, increasing the total acquisition latency, but reducing the latency from sample capture to the initiation of the tracking process. In this case, stage 2 was configured to perform a Doppler search of ±10 Hz around the Doppler frequency detected in Stage 1. In this way, stage 2 reduces the time required to transition from sample capture to tracking down to 1.5 s. This ensures that when the receiver begins tracking the detected signals, the Doppler frequency of the detected signals remains close to the Doppler frequency estimated during acquisition. This approach is expected to increase the probability of successfully tracking the detected signals.

The receiver was tested using the configuration shown in Table 13, successfully acquiring and tracking weak Galileo E1b/c signals and detecting the telemetry preambles. However, we did not perform tests to quantify the impact of enabling or disabling the second acquisition stage when using the OCXO. Detailed lists of weak signals that were successfully acquired and tracked by the receiver, utilizing an OCXO, are provided in Section 4.3.

Due to the manual estimation and correction of the CFO induced by the TCXOs being inconvenient, we switched to using an OCXO, configuring the receiver as outlined in Table 13 for subsequent tests. Nevertheless, it would be desirable to improve the acquisition algorithm to eliminate the need for manual CFO compensation. Achieving this would make the use of TCXOs practical in the proposed receiver, as TCXOs enable the detection of weak GNSS signals with a $C/N_{0}$ as low as 20 dB-Hz.

### 4.3. Acquisition and Tracking of GNSS Signals in Real Time

The objective of the acquisition and tracking tests was to confirm the receiver’s capability to process GNSS signals and derive navigation solutions in real-time. The receiver underwent testing using the test setup described in Section 4.1. We assessed the performance of the receiver using signals received at nominal power and using weak signals with a $C/N_{0}$ down to 20 dB-Hz. The OCXO was used for these tests. The test results presented in this section were obtained using the GESTALT testbed rooftop GNSS antenna facility [46]. Using the Tallysman TW8825 active antenna and a variable attenuator in outdoor conditions could yield similar results. Each test was conducted over a period of 10 min.

The test procedure is outlined in Table 14. Steps 1 and 2 verify the receiver’s ability to process Galileo E1b/c signals in high-sensitivity mode concurrently with Galileo E5a and GPS signals in normal-sensitivity mode. Steps 3 and 4 confirm the receiver’s ability to process Galileo E1b/c signals at a $C/N_{0}$ down to 20 dB-Hz. The OCXO was used for these tests.

#### 4.3.1. Tests Results Using GPS and Galileo Signals at Nominal Power Levels

Table 15 shows the test results obtained using signals received at nominal power. The receiver successfully acquired and tracked Galileo E1b/c, Galileo E5a, GPS L1 C/A, and GPS L5 signals, achieving navigation solutions with TTFF values of about 1 minute.

#### 4.3.2. Tests Results Using Weak Galileo E1b/c Signals

The test results using weak signals are summarized in Table 16. The receiver successfully acquired and tracked weak Galileo E1b/c signals, including those with a $C/N_{0}$ down to 20 dB-Hz, obtaining navigation solutions with TTFF values ranging from 1 to 2 min. The receiver used the observations obtained from signals with a $C/N_{0}$ at or close to 20 dB-Hz to compute the navigation solutions.

The results demonstrate that the receiver can detect weak Galileo E1b/c signals with a $C/N_{0}$ as low as 20 dB-Hz in real-time and obtain navigation solutions. However, while many measurements yielded a TTFF between 1 and 2 min, there were instances where the receiver faced challenges in determining position using the observables derived from weak signals.

### 4.4. Precision of the Navigation Solutions

The authors evaluated the quality of navigation solutions obtained by the receiver in high-sensitivity mode through two tests. In the first test, the receiver processed Galileo E1b/c signals in high-sensitivity mode along with Galileo E5a and GPS signals in normal-sensitivity mode, all received at nominal power levels. In the second test, the receiver processed weak Galileo E1b/c signals with a $C/N_{0}$ of approximately 20 dB-Hz. The measurements were conducted using the GESTALT testbed rooftop antenna [46]. The experimental scenario described in Section 4.1 was employed, along with the test setup depicted in Figure 10. The OCXO was used for these tests.

The quality of the navigation solutions was assessed using standard positioning precision measurements and corresponding static confidence regions. Precision, which indicates how closely a solution aligns with the mean of all obtained solutions, reflects repeatability or the spread of the measurement. Key confidence measurements include Distance Root Mean Square (DRMS) and Circular Error Probability (CEP) for 2D positioning, as well as Spherical Accuracy Standard (SAS), Mean Radial Spherical Error (MRSE), and Spherical Error Probable (SEP) for 3D positioning. Formulas for 2D and 3D position confidence regions are provided in Table 17 and Table 18, respectively, [49,50].

Similar to the measurements reported in [21], the receiver’s latitude, longitude, and height coordinates were converted into a local East–North–Up (ENU) coordinate system, with the WGS-84 reference ellipsoid as the reference. Standard deviations for the East (E), North (N), and Up (U) coordinates were computed. Equation (Equation 9) demonstrates the calculation of the standard deviation for East coordinates, denoted as E[n], using the mean value $\bar{E}$ of the East coordinates and the number of position fixes, *L*. The standard deviation calculations for North (N) and Up (U) coordinates follow the same procedure as Equation (Equation 9) for East coordinates.
(9)$$\sigma_{E}^{\left(precision\right)}=\sqrt{\frac{1}{L-1}\sum_{l=1}^{L}{\left(E\left[n\right]-\bar{E}\right)}^{2}}\;\;\;[m]$$

The receiver processed GNSS signals in real time and dumped PVT solutions to a file, configured for single-point positioning mode. Each test was conducted over a period of 10 min.

#### 4.4.1. Tests Results Using GPS and Galileo Signals at Nominal Power Levels

The 2D precision results are presented in Table 19, and the 3D precision results are provided in Table 20. The receiver successfully processed 7 Galileo E1b/c, 7 Galileo E5a, 7 GPS L1 C/A, and 4 GPS L5 signals. The navigation solutions were obtained using up to 11 observations. The satellites broadcasting the unhealthy flag were not used for the computation of the PVT.

Significantly, 2D precision measurements demonstrate greater accuracy compared to their 3D counterparts, as expected due to the superior GNSS accuracy in the horizontal plane when contrasted with the vertical plane. This variation is influenced by the angle between the line of sight to various satellites and the Earth’s surface.

#### 4.4.2. Tests Results Using Weak Galileo E1b/c Signals

The accuracy of the navigation solutions was evaluated using weak signals. These measurements yielded variable outcomes. The test results shown below are an example of the performance that could be achieved.

The $C/N_{0}$ of the received Galileo E1b/c signals was decreased to the levels shown in Table 21 using the variable attenuator. The receiver processed the received signals and obtained the navigation solutions using four observations. The 2D precision results are presented in Table 22, and the 3D precision results are provided in Table 23.

The precision of navigation solutions can significantly decrease when tracking weak GNSS signals, and the quality of these solutions becomes highly variable, while the results in Table 22 and Table 23 are achievable, errors can increase depending on satellite geometry and signal strength. Implementing algorithms to enhance the accuracy of PVT solution when tracking severely degraded signals is a focus for future work.

### 4.5. Estimated Power Consumption

The estimated power consumption of the XCZU9EG-FFVB1156-2-e SoC-FPGA, when operating a high-sensitivity GNSS receiver, is 12 W. This estimate, provided by FPGA design tools, assumes an average processor load of 75%. Such a load occurs when the receiver processes a large volume of GNSS signals simultaneously, including 10 signals each from Galileo E1, GPS L1 C/A, Galileo E5a, and GPS L5. In contrast, in [21], the authors tested the GNSS-SDR software receiver on a personal computer (PC) with a 28 W 11th generation Intel Core i7-1185G7 processor at 3 GHz. Processing 26 channels in real-time across multiple bands and constellations resulted in a similar processor load. This comparison suggests that the SoC-FPGA implementation is capable of processing over 50% more signals in real time within the same power budget.

To assess the average processor load when processing multiple signals, the authors connected the receivers to the GESTALT testbed’s rooftop GNSS antenna facility [46]. The receivers were allowed to simultaneously track the same signals across multiple channels, to ensure that every channel was tracking at least one satellite signal. We estimated the average processor load using the receiver’s operating system tools.

## 5. Conclusions

This paper reports the design, proof-of-concept implementation, and preliminary performance assessment of a HS-GNSS receiver, providing practical details of a working prototype. The proposed design is based on a SoC-FPGA GNSS receiver architecture, combining the massive parallelism and energy efficiency of FPGAs with the flexibility of embedded processors. The proof-of-concept demonstrator is implemented using COTS products.

This receiver can be used to test novel high-sensitivity algorithms, including those addressing typical effects of indoor GNSS scenarios. The device is fully configurable, delivering GNSS products in standard open formats, including GPX, KML, GeoJSON, RINEX files (for navigation and observables), RTCM v3.2, and NMEA-0183 for sensor integration.

After presenting the system architecture, the receiver was tested. It demonstrated the capability to acquire and track weak Galileo E1b/c signals in real-time. The receiver processes Galileo E1b/c signals with a carrier-to-noise ratio ($C/N_{0}$) as low as 20 dB-Hz, successfully obtaining navigation solutions. Moreover, it simultaneously handles Galileo E5a, GPS L1 C/A, and GPS L5 signals at their nominal power levels. This capability enhances the availability of satellite signals wherever possible.

The estimated power consumption of the SoC-FPGA implementing the proposed receiver is 12 W, enabling its operation in battery-powered devices. The physical dimensions of the receiver prototype are relatively large, measuring 23.749 cm × 24.384 cm, with the analog front-end measuring 14 cm × 9 cm. However, the receiver remains portable. It can be placed in backpacks, for instance, to facilitate testing in soft indoor environments.

In the context of future work, we envision several enhancements for the proposed receiver. At present, incorporating a TCXO into the receiver requires a manual estimation and correction process to address the CFO resulting from clock instability. However, it is desirable to increase the robustness of the receiver algorithms in this regard. Future developments include enhancing the acquisition algorithm’s resilience to CFO and integrating it with the proposed receiver utilizing a TCXO.

Furthermore, the acquisition and tracking of weak GNSS signals are currently limited to Galileo E1b/c. Receiver availability in moderate indoor scenarios (the proportion of time the software receiver is in a functioning condition) may be significantly improved by acquiring and tracking weak GPS and Galileo E5a signals as well. This would improve the success rate of positioning and the quality of the navigation solutions by receiving more satellite signals in harsh environments.

Finally, the proposed receiver is primarily designed to acquire and track weak GNSS signals. The quality of navigation solutions, however, can significantly deteriorate when tracking these signals due to variations in received signal strength, satellite geometry, and potential signal reflections. Currently, the receiver does not incorporate multipath mitigation algorithms, which are relevant for addressing these challenges. Therefore, implementing such algorithms and improving the accuracy of navigation solutions through improved methods is a key area for future work.

## Figures and Tables

**Figure 1 sensors-24-01416-f001:**
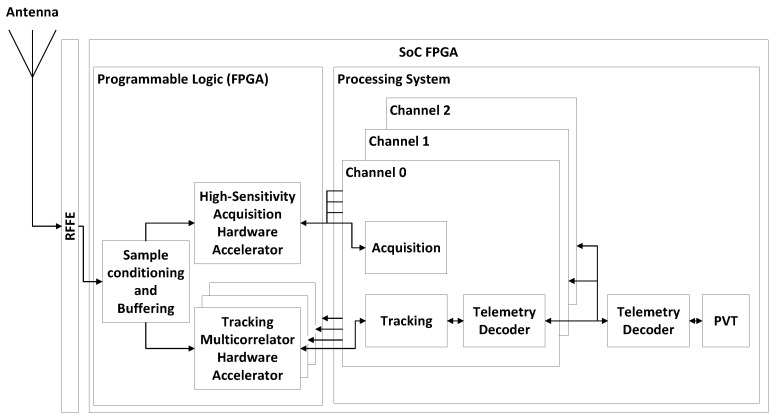
Block diagram of the SoC-FPGA GNSS receiver architecture.

**Figure 2 sensors-24-01416-f002:**
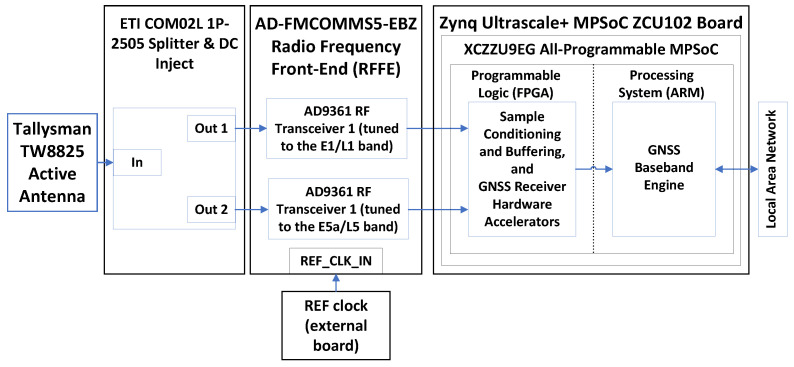
High-sensitivity GNSS receiver block diagram.

**Figure 3 sensors-24-01416-f003:**
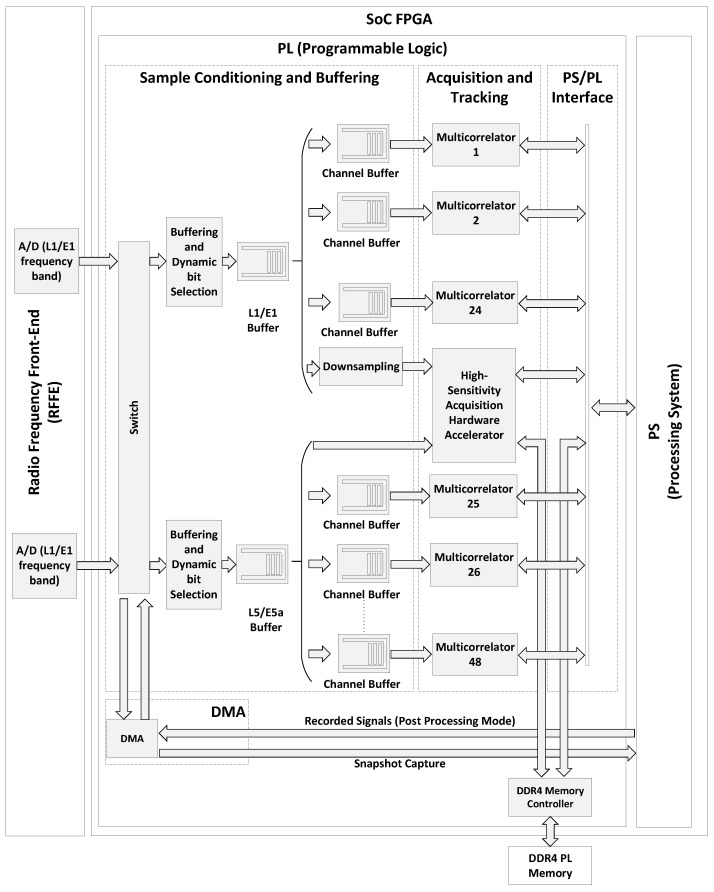
Detailed FPGA design.

**Figure 4 sensors-24-01416-f004:**
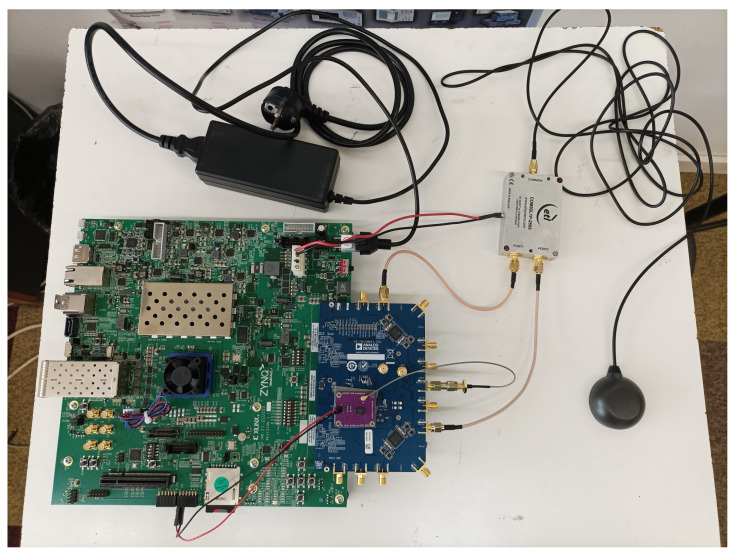
Picture of the high-sensitivity GNSS receiver prototype.

**Figure 5 sensors-24-01416-f005:**
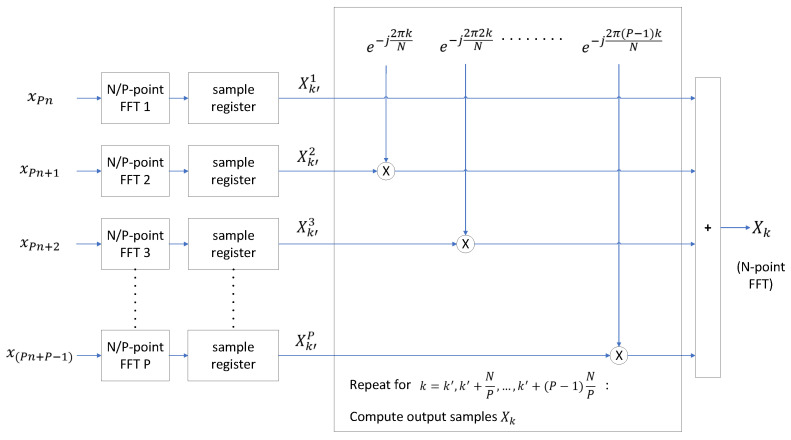
Computation of an *N*-point FFT as a combination of *P* FFTs, each of N/P points.

**Figure 6 sensors-24-01416-f006:**
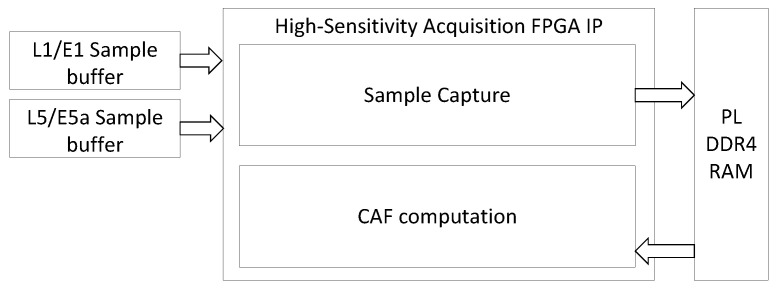
High-sensitivity acquisition FPGA hardware accelerator block diagram.

**Figure 7 sensors-24-01416-f007:**
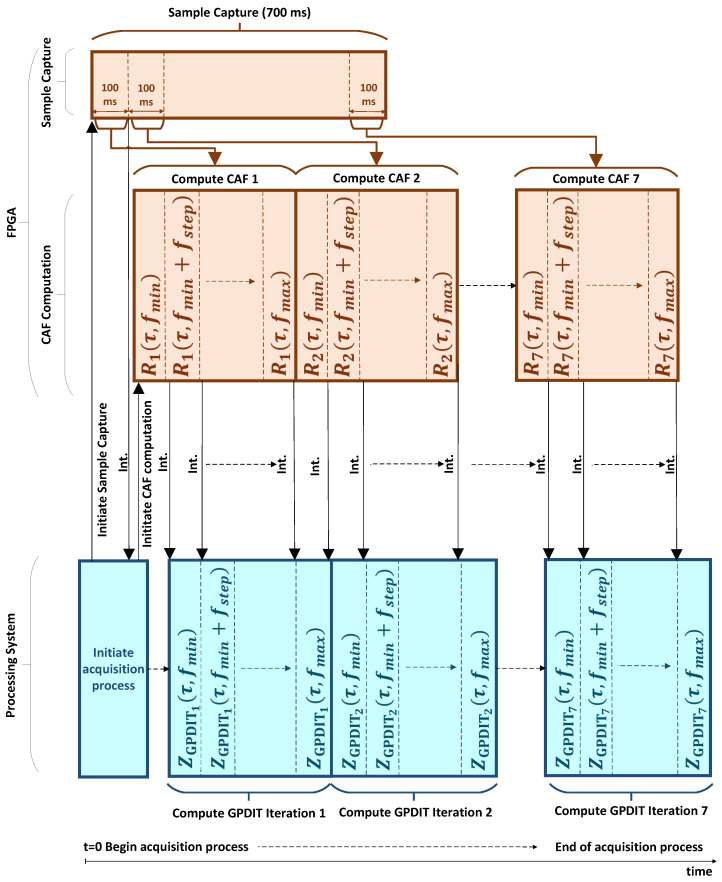
Acquisition timing diagram.

**Figure 8 sensors-24-01416-f008:**
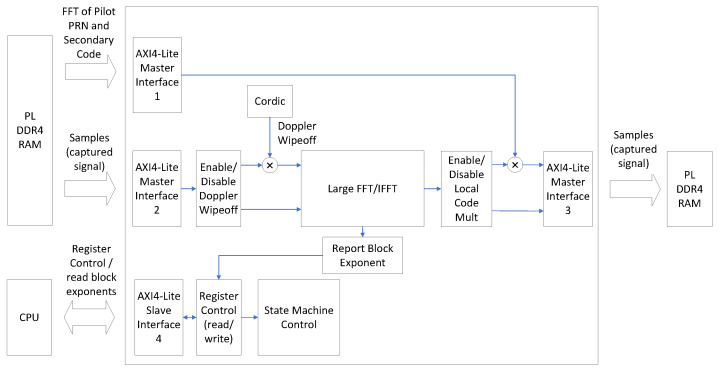
CAF computation module.

**Figure 9 sensors-24-01416-f009:**
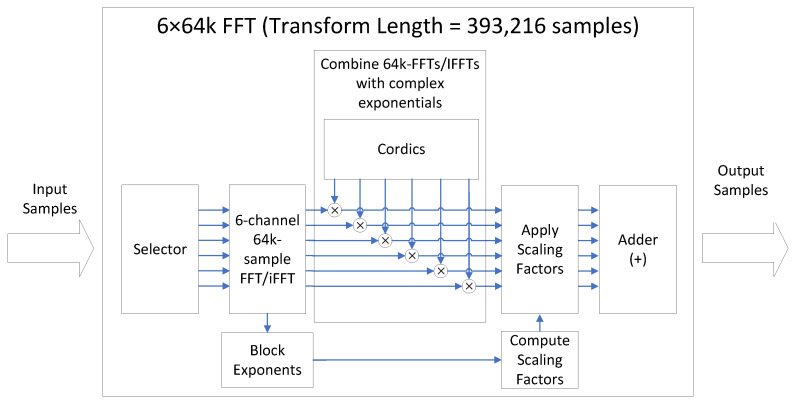
Large FFT/iFFT module implemented in the high-sensitivity acquisition.

**Figure 10 sensors-24-01416-f010:**
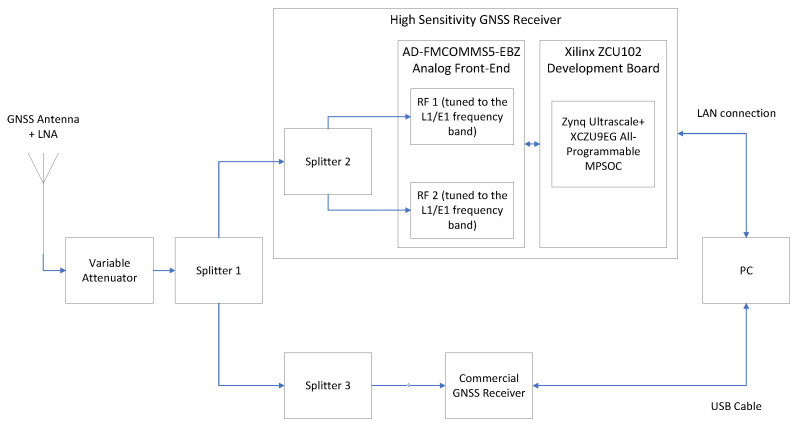
Test setup.

**Table 1 sensors-24-01416-t001:** Default receiver operating modes.

Signal	Operating Mode	Receiver Sensitivity
Galileo E1b/c	High-Sensitivity mode	20 dB-Hz
GPS L1 C/A GPS L5 Galileo E5a	Normal-sensitivity mode	approx. 37 dB-Hz

**Table 2 sensors-24-01416-t002:** Assistance data.

Assistance Data	Source
Reference date and time	Receiver embedded OS clock
Reference user location	GNSS-SDR configuration file
Galileo ephemeris data Galileo ionospheric data Galileo UTC model	XML files

**Table 3 sensors-24-01416-t003:** Purpose of the assistance data.

Objective	Assistance Data Used
Doppler frequency estimation	Reference date and time, reference user location, Galileo ephemeris data
TOW estimation	Reference date and time
Computation of the navigation solutions	Galileo ephemeris data, ionospheric data, and UTC model

**Table 4 sensors-24-01416-t004:** Input parameters of the high-sensitivity acquisition.

Input Parameter	Definition
$f_{min}$	Minimum tested Doppler frequency
$f_{max}$	Maximum tested Doppler frequency
$f_{step}$	Doppler search step
C[k]	FFT of the Galileo pilot signal to be detected, with a tiered code duration of 100 ms

**Table 5 sensors-24-01416-t005:** Output parameters of the high-sensitivity acquisition.

Output Parameter	Definition
$f_{d_{acq}}$	Estimated Doppler frequency of the received signal
$\tau_{acq}$	Estimated code phase

**Table 6 sensors-24-01416-t006:** Variables introduced in the description of the high-sensitivity algorithm.

Variable	Definition
$x_{IN}\left[n\right]$	Received GNSS signal input sample stream
${\hat{P}}_{IN}$	Input signal power estimation
$n_{nc}$	Current non-coherent combination number
$N_{nc}$	Maximum number of non-coherent combinations. This value is set to 7
*N*	Number of samples used for the coherent integration, representing 100 ms
$\left[f_{min},f_{max}\right]$	Doppler frequency span
$f_{d}$	Tested Doppler frequency
$T_{s}$	Sampling period
$R_{k}\left(\tau ,f_{d}\right)$	*k*-th successive Cross-ambiguity function (CAF)
$Z_{\mathrm{GPDIT}}\left(\tau ,f_{d}\right)$	Result of the NCI using the GPDIT strategy as shown in Equation (Equation 3) [7]
$\Gamma_{\mathrm{GLRT}}$	Generalized Likelihood Ratio Test (GLRT) function with normalized variance

**Table 7 sensors-24-01416-t007:** Acquisition process in two steps.

Stage	Algorithm
Stage 1	Capture samples from the analog front-end. Perform acquisition using a Doppler search space around the predicted Doppler frequency (the Doppler search space is large enough to account for any CFO and Doppler prediction inaccuracy).
Stage 2	Capture samples from the analog front-end. Perform acquisition using a very small Doppler search space around the Doppler frequency estimated during stage 1. As a result, when executing stage 2, the system quickly transitions from capturing samples to starting the tracking process.

**Table 8 sensors-24-01416-t008:** Tracking multicorrelator configuration.

Signal	Configuration
GPS L1 C/A	E, P, L
Galileo E1b/c	VE, E, P, L, VL (Pilot Component)
	P (Data Component)
GPS L5	E, P, L (Pilot Component)
	P (Data Component)
Galileo E5a	E, P, L (Pilot Component)
	P (Data Component)

**Table 9 sensors-24-01416-t009:** Acquisition latency: test procedure.

Step	Description
Step 1	Configure the Doppler search space of the two-stage acquisition process.
Step 2	Apply weak Galileo E1b/c signals to the receiver, aiming for a $C/N_{0}$ ratio that is approximately 20 dB-Hz
Step 3	Verify that the receiver acquires and initiates tracking of the weak signals, successfully detecting the telemetry preambles. Measure the acquisition latency.

**Table 10 sensors-24-01416-t010:** CFO in the E1 frequency band when using a TXCO.

TCXO	Measured CFO in the E1 Frequency Band
TCXO 1	−61 Hz to 89 Hz
TCXO 2	109 Hz to 156 Hz
TCXO 2	−273 Hz to −185 Hz

**Table 11 sensors-24-01416-t011:** Acquisition latency using a TCXO.

Stage	Doppler Search Space	Measured Latency
Stage 1	±100 Hz around the Doppler frequency predicted using assistance data, in steps of 5 Hz	initial part of sample capture: 100 ms 7 non-coherent combinations: 14 s Total: 14.1 s
Stage 2	±15 Hz around the Doppler frequency estimated in step 1, in steps of 5 Hz	initial part of sample capture: 100 ms 7 non-coherent iterations: 2 s Total: 2.1 s

**Table 12 sensors-24-01416-t012:** List of signals acquired and tracked when utilizing a TCXO.

TCXO	E1b/c Signals	$C/N_{0}$ in dB-Hz
TCXO 1	E1	33
	E4	27
	E13	30
	E15	20
	E21	33
	E26	27
TCXO 2	E2:	31
	E11:	28
	E18:	27
	E24:	21
	E25:	31
	E36:	36
TCXO 3	E10:	29
	E12:	25
	E24:	26
	E25:	23
	E33:	22

**Table 13 sensors-24-01416-t013:** Acquisition latency using an OCXO.

Stage	Doppler Search Space	Measured Latency
Stage 1	±50 Hz around the Doppler frequency predicted using assistance data, in steps of 5 Hz	Initial part of sample capture: 100 ms 7 non-coherent iterations: 7 s Total: 7.1 s
Stage 2	±10 Hz around the Doppler frequency estimated in step 1, in steps of 5 Hz	Initial part of sample capture: 100 ms 7 non-coherent iterations: 1.4 s Total: 1.5 s

**Table 14 sensors-24-01416-t014:** Test procedure.

Step	Description
Step 1	Use the receiver in normal-sensitivity mode to obtain the ephemeris data, the ionospheric data, and the UTC model for the visible Galileo satellites.
Step 2	Use the receiver in high-sensitivity mode with assistance data, including the data obtained in step 1, to receive GNSS signals at nominal power and obtain navigation solutions
Step 3	Stop the receiver and increase the signal attenuation until the $C/N_{0}$ of some Galileo E1b/c signals is down to 20 dB-Hz
Step 4	Use the receiver in high-sensitivity mode, to acquire and track weak GNSS signals and obtain navigation solutions.

**Table 15 sensors-24-01416-t015:** Processing GNSS signals at nominal power levels.

Test	Signals Tracked	TTFF
Test 1	8 Galileo E1b/c + 8 Galileo E5a + 7 GPS L1 C/A + 5 GPS L5	1 min 03 s
Test 2	8 Galileo E1b/c + 8 Galileo E5a + 6 GPS L1 C/A + 4 GPS L5	57 s
Test 3	8 Galileo E1b/c + 8 Galileo E5a + 7 GPS L1 c/a + 4 GPS L5	1 min 12 s

**Table 16 sensors-24-01416-t016:** Processing weak Galileo E1b/c signals.

Test	E1b/c Signals Tracked and $C/N_{0}$ in dB-Hz	TTFF
Test 1	E2: 27 dB-Hz	1 min 40 s
	E8: 20 dB-Hz	
	E10: 20 dB-Hz	
	E25: 25 dB-Hz	
	E36: 23 dB-Hz	
Test 2	E13: 23 dB-Hz	53 s
	E15: 28 dB-Hz	
	E21: 20 dB-Hz	
	E27: 28 dB-Hz	
	E30: 26 dB-Hz	
	E34: 26 dB-Hz	
Test 3	E3: 21 dB-Hz	1 min 15 s
	E8: 22 dB-Hz	
	E13: 22 dB-Hz	
	E15: 21 dB-Hz	

**Table 17 sensors-24-01416-t017:** Most common 2D precision measures.

Measure	Formula	Confidence Region Probability
2D 2DRMS	$2\sqrt{\sigma_{E}^{2}+\sigma_{N}^{2}}$	95%
2D DRMS	$\sqrt{\sigma_{E}^{2}+\sigma_{N}^{2}}$	65%
2D CEP	$0.62\sigma_{N}+0.56\sigma_{E}$ (accurate if $\frac{\sigma_{N}}{\sigma_{E}}>0.3$)	50%

**Table 18 sensors-24-01416-t018:** Most common 3D precision measures.

Measure	Formula	Confidence Region Probability
3D 99% SAS	$1.122(\sigma_{E}+\sigma_{N}+\sigma_{U})$	99%
3D 90% SAS	$0.833(\sigma_{E}+\sigma_{N}+\sigma_{U})$	90%
3D MRSE	$\sqrt{\sigma_{E}^{2}+\sigma_{N}^{2}+\sigma_{U}^{2}}$	61%
3D SEP	$0.51(\sigma_{E}+\sigma_{N}+\sigma_{U})$	50%

**Table 19 sensors-24-01416-t019:** The 2D precision results.

Measure	Results [m]	Confidence Region Probability
2D 2DRMS	4.3	95%
2D DRMS	2.1	65%
2D CEP	1.8	50%

**Table 20 sensors-24-01416-t020:** The 3D precision results.

Measure	Results [m]	Confidence Region Probability
3D 99% SAS	8.5	99%
3D 90% SAS	6.3	90%
3D MRSE	5.0	61%
3D SEP	3.8	50%

**Table 21 sensors-24-01416-t021:** Galileo satellites tracked and $C/N_{0}$.

Satellite	$C/N_{0}$ in dB-Hz
E3	20 dB-Hz
E8	23 dB-Hz
E13	22 dB-Hz
E15	20 dB-Hz

**Table 22 sensors-24-01416-t022:** The 2D precision results using weak Galileo E1b/c signals.

Measure	Results [m]	Confidence Region Probability
2D 2DRMS	7.5	95%
2D DRMS	3.8	65%
2D CEP	3.1	50%

**Table 23 sensors-24-01416-t023:** The 3D precision results using weak Galileo E1b/c signals.

Measure	Results [m]	Confidence Region Probability
3D 99% SAS	18.0	99%
3D 90% SAS	13.4	90%
3D MRSE	11.4	61%
3D SEP	8.9	50%

## Data Availability

The GNSS-SDR software receiver is accessible online [26]. For those seeking the high-sensitivity version of the GNSS-SDR receiver, it can be obtained by reaching out to the authors directly. The source code for the FPGA implementation is proprietary.

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
