# Peer review of "Implementation of a High-Sensitivity Global Navigation Satellite System Receiver on a System-on-Chip Field-Programmable Gate Array Platform"

_sensors, 2024, doi:10.3390/s24051416_

Round 1
Reviewer 1 Report
Comments and Suggestions for Authors
The manuscript presents a system-level programmable logic gate array (SoC-FPGA) architecture for prototyping high-sensitivity global navigation satellite system (GNSS) receivers. The authors claim that the proposed architecture combines the advantages of software-defined radio (SDR) technology and FPGA’s energy efficiency, enabling the development of compact, portable, multi-channel, multi-constellation GNSS receivers that can test novel and non-standard GNSS features and real-time signals. The authors describe the proposed architecture and design methodology, review the practical applications of space GNSS receivers and GNSS repeaters, and introduce a generic GNSS receiver design and preliminary performance evaluation as a testbed for future research. The manuscript mostly briefly mentions some theoretical issues without delving into thorough analysis or solutions. The topic is relevant and interesting for the journal’s scope and audience. The proposed architecture and design methodology are novel and innovative, and the results are convincing and significant. The manuscript has several issues, such as:
1) The authors should provide more details on the performance evaluation of the receiver, such as the data sources, the data formats, the data processing, the metrics, and the comparison methods. This would help the readers to better assess the validity and the significance of the results.
2) In Section 2, the author mentioned that the design and optimization of signal acquisition and processing algorithms for GNSS receivers are important directions, but specific algorithms or implementation details were not provided.
3) This article effectively discusses the definition, impact, and estimation methods of residual frequency error, but lacks detailed analysis or validation of its performance and effects in practical scenarios. For example, the author does not provide the numerical range or distribution of residual frequency error, nor does the author compare the results of different estimation methods under varying signal-to-noise ratios or signal strengths. Additionally, specific impacts or improvements of residual frequency error on positioning accuracy are not presented. Therefore, I believe there is significant room for improvement in the evaluation of residual frequency error in this article, and more data and experiments are needed to support its conclusions.
4) The authors should provide more details on the hardware and software components of the proposed architecture, such as the specifications of the SoC-FPGA device, the operating system, the programming languages, the libraries, and the tools used for development and testing. Could the author provide more detail to help readers to better understand the design choices and the trade-offs involved in the implementation.
5) The authors should give more explanation about how they handle the synchronization and communication between the hardware logic and the embedded processor in the SoC-FPGA device, as well as between the receiver and the external devices, such as the antenna, the data storage, and the user interface. This would help the readers to better appreciate the challenges and solutions in the system integration.
6) The authors should provide more details on the generic GNSS receiver design, such as the signal processing algorithms, the data structures, the parameters, and the outputs. This would help the readers to better follow the functionality and the performance of the receiver.
Comments on the Quality of English Language
No comment.
Reviewer 2 Report
Comments and Suggestions for Authors
This paper reports the design, proof-of-concept implementation, and preliminary performance assessment of a low-cost real-time high-sensitive GNSS Receiver in weak signal conditions. The research topic is great of significance. My main concerns are listed as follows:
1. In the line 158 of Page 4, “Assistance (A-GNSS) is used to speed up….”. However, in Page 32, A-GNSS stands for Assisted GNSS. The authors should make the A-GNSS clearly.
2. In Table 2 of Page 10, what does the ~20 dB-Hz means? Does it mean almost 20 dB-Hz? Please make it more clear.
3. In the line 451 of Page 12, “K-th” contains the variable k, which should italic. The authors should check this paper carefully and solve all kinds of these problems.
4. In the equation (2), what does the Z_x represent?
5. In Algorithm 1, what are the input parameters? What are the output parameters? Please make it more clear.
6. In weak signal conditions, including urban canyons and indoor scenarios, there are obviously many reflected or refracted GNSS signals. The received signals should be a combination of direct, reflected or refracted GNSS signals. How to consider these multipath effects?
7. In Section 4, all of results are presented by Tables. It would be better if the authors can provide some Figures.
Comments on the Quality of English Language
Moderate editing of English language required.
Reviewer 3 Report
Comments and Suggestions for Authors
Editor:
This paper presents a high-sensitivity GNSS receiver on a Soc-FPGA platform. At the same time, through theoretical derivation and targeted experiments, the feasibility of the designed highly sensitive GNSS receiver is demonstrated. Capable of acquisition and tracking Galileo E1b/c signals with a C/N0 down to 20dB-Hz. The work in the article is partly innovative but the details still need to be further improved. I have the following comments to authors.
1. The abstract lacks clarity in summarizing the innovation and contribution of the article, and the logical expression of the language requires refinement.
2. In section 1.2, the contribution statement of the article should be succinctly and precisely summarized.
3. Beyond line 247 in Section 2, the connection between the retrospective analysis of the acquisition algorithm and its relevance to this paper lacks clarity. There is no explicit indication of the guiding significance of this research to the present study. The section's review of the research background on the acquisition algorithm might be more appropriately placed in the introduction rather than as a standalone section, as its independent significance is not distinctly evident.
4. In figure 2, the module name "high-sensitivity acquisition" lacks clarity in its meaning. In reality, high-sensitivity processing is not implemented for signals at L1/L5/E5 frequencies; only the E1 frequency supports high-sensitivity processing. The term "high-sensitivity acquisition" in the figure refers to two distinct operational modes, and the expression is not precise enough.
5. In section 3.6, the specific implementation details of high-sensitivity acquisition lack detailed description. It is unclear whether the pilot and data channels are simultaneously tracked and how the 100ms tracking is achieved.
6. In table 7, the distinction among the three sets of comparisons regarding TCXO is not clearly articulated. An explanation is needed for the significant differences observed in the results of the three test sets.
7. In line 880, the article states that no CFO correction is needed when using OCXO. The significance of employing a two-stage acquisition process remains unclear. Additionally, there is a lack of empirical data support for the results after CFO correction.
8. Section 4.3 lacks a description of the experimental scenario. Please provide additional details for Section 4.3.
9. The abstract mentions the implementation of low power consumption, but in Section 4.4, the power consumption estimation analysis fails to provide a comparison supporting this conclusion.
Comments on the Quality of English Language
Editor:
This paper presents a high-sensitivity GNSS receiver on a Soc-FPGA platform. At the same time, through theoretical derivation and targeted experiments, the feasibility of the designed highly sensitive GNSS receiver is demonstrated. Capable of acquisition and tracking Galileo E1b/c signals with a C/N0 down to 20dB-Hz. The work in the article is partly innovative but the details still need to be further improved. I have the following comments to authors.
1. The abstract lacks clarity in summarizing the innovation and contribution of the article, and the logical expression of the language requires refinement.
2. In section 1.2, the contribution statement of the article should be succinctly and precisely summarized.
3. Beyond line 247 in Section 2, the connection between the retrospective analysis of the acquisition algorithm and its relevance to this paper lacks clarity. There is no explicit indication of the guiding significance of this research to the present study. The section's review of the research background on the acquisition algorithm might be more appropriately placed in the introduction rather than as a standalone section, as its independent significance is not distinctly evident.
4. In figure 2, the module name "high-sensitivity acquisition" lacks clarity in its meaning. In reality, high-sensitivity processing is not implemented for signals at L1/L5/E5 frequencies; only the E1 frequency supports high-sensitivity processing. The term "high-sensitivity acquisition" in the figure refers to two distinct operational modes, and the expression is not precise enough.
5. In section 3.6, the specific implementation details of high-sensitivity acquisition lack detailed description. It is unclear whether the pilot and data channels are simultaneously tracked and how the 100ms tracking is achieved.
6. In table 7, the distinction among the three sets of comparisons regarding TCXO is not clearly articulated. An explanation is needed for the significant differences observed in the results of the three test sets.
7. In line 880, the article states that no CFO correction is needed when using OCXO. The significance of employing a two-stage acquisition process remains unclear. Additionally, there is a lack of empirical data support for the results after CFO correction.
8. Section 4.3 lacks a description of the experimental scenario. Please provide additional details for Section 4.3.
9. The abstract mentions the implementation of low power consumption, but in Section 4.4, the power consumption estimation analysis fails to provide a comparison supporting this conclusion.
Reviewer 4 Report
Comments and Suggestions for Authors
This paper presents a comprehensive introduction to the implementation of a high-sensitivity GNSS receiver on an FPGA platform. The system architecture is designed for two operating modes: high-sensitivity and normal-sensitivity. The paper emphasizes the high-sensitivity aspects of the system design and provides descriptions of the normal-sensitivity algorithms, data decoding, and navigation solution computation to ensure the integrity of the receiver design. Using an OCXO, the authors conducted real-time tests on the acquisition, tracking, and navigation performance of the proposed high-sensitivity GNSS receiver system. The results validate that the system successfully outputs positioning results with E1b/c signals at a low C/N0 value of approximately 20~23 dB-Hz.
This paper is well-organized, presenting methodologies clearly and coherently. The details of the receiver design are articulated in a way that is easy for readers to follow. Additionally, the chosen topic aligns well with the special issue of the journal. I think it can be accepted in its current form.
I just have a few minor comments and some questions out of curiosity:
1. Regarding the hardware accelerator (Section 3.4.2), how the length of P is determined to make a trade-off between hardware resource reduction and acquisition performance for Doppler uncertainty is not discussed. Were specific strategies employed in the system design to decide an optimal number of P FFTs?
2. Are the DLLs and PLLs/FLLs (including their orders and bandwidths) involved in the high-sensitivity tracking mode? Or is an open loop utilized in this mode? It might be beneficial to briefly address and include this point in Section 3.6.1 for clarity.
Again, I appreciate the authors’ great work.
Round 2
Reviewer 1 Report
Comments and Suggestions for Authors
No further comment, this paper is fine to me.
Reviewer 3 Report
Comments and Suggestions for Authors
This paper presents a high-sensitivity GNSS receiver on a Soc-FPGA platform. At the same time, through theoretical derivation and targeted experiments, the feasibility of the designed highly sensitive GNSS receiver is demonstrated. Capable of acquisition and tracking Galileo E1b/c signals with a C/N0 down to 20dB-Hz. After revision, the article's logic became clearer and the experimental description became more specific.